# Unleashing a novel function of Endonuclease G in mitochondrial genome instability

**Sumedha Dahal[1], Humaira Siddiqua[1], Shivangi Sharma[1], Ravi K Babu[1], Diksha Rathore[1], Sheetal Sharma[2], Sathees C Raghavan[1]***

[1]Department of Biochemistry, Indian Institute of Science Bangalore, Bangalore, India; [2]Department of Experimental Medicine and Biotechnology, Post Graduate Institute of Medical Education and Research, Chandigarh, India

**Abstract** Having its genome makes the mitochondrion a unique and semiautonomous organelle within cells. Mammalian mitochondrial DNA (mtDNA) is a double-stranded closed circular molecule of about 16 kb coding for 37 genes. Mutations, including deletions in the mitochondrial genome, can culminate in different human diseases. Mapping the deletion junctions suggests that the breakpoints are generally seen at hotspots. '9 bp deletion' (8271–8281), seen in the intergenic region of cytochrome c oxidase II/tRNA$^{Lys}$, is the most common mitochondrial deletion. While it is associated with several diseases like myopathy, dystonia, and hepatocellular carcinoma, it has also been used as an evolutionary marker. However, the mechanism responsible for its fragility is unclear. In the current study, we show that Endonuclease G, a mitochondrial nuclease responsible for nonspecific cleavage of nuclear DNA during apoptosis, can induce breaks at sequences associated with '9 bp deletion' when it is present on a plasmid or in the mitochondrial genome. Through a series of in vitro and intracellular studies, we show that Endonuclease G binds to G-quadruplex structures formed at the hotspot and induces DNA breaks. Therefore, we uncover a new role for Endonuclease G in generating mtDNA deletions, which depends on the formation of G4 DNA within the mitochondrial genome. In summary, we identify a novel property of Endonuclease G, besides its role in apoptosis and the recently described 'elimination of paternal mitochondria during fertilisation.

*For correspondence:
sathees@iisc.ac.in

**Competing interest:** The authors declare that no competing interests exist.

## Editor's evaluation

This manuscript is of interest to researchers in the field of mitochondrial genome stability and mitochondrial genetic diseases and reports valuable findings that convincingly demonstrate that endonuclease G preferentially binds to mitochondrial genome regions which have a potential for forming G4 tetraplexes and induces DNA breaks that may lead to a common 9 bp deletion in the mitochondrial genome.

## Introduction

The mitochondria are semiautonomous organelles with their genome. The mammalian mitochondrial genome encodes for 13 respiratory chain proteins, 22 tRNAs and 2 rRNAs (*Chen and Butow, 2005*; *Clayton, 1984*; *Holthöfer et al., 1999*). Due to its vital role in oxidative phosphorylation, mitochondrial DNA is frequently exposed to reactive oxygen species (ROS) and is more prone to DNA damage than nuclear DNA, leading to the accumulation of 50 times more mutations, including deletions (*Chen et al., 2011*; *Hudson et al., 1998*; *Michikawa et al., 1999*; *Pakendorf and Stoneking, 2005*; *Yakes and Van Houten, 1997*). Mitochondrial mutations have been associated with ageing and several

disease conditions like myopathies, dystonia, cancer etc. (*Chen et al., 2011*; *Taylor and Turnbull, 2005*). These mutations include point mutations, mismatches, or deletions, which affect the coding of essential proteins involved in oxidative phosphorylation and the respiratory chain (*Penta et al., 2001*). Mitochondrial mutations, especially large-scale deletions of 200–5000 bp, were associated with the progression of breast, colorectal, renal, and gastric cancers (*Bianchi et al., 2001*; *Carew and Huang, 2002*; *Modica-Napolitano et al., 2007*; *Penta et al., 2001*). However, the correlation between mtDNA deletions and cancer remains unsettled. The well-studied phenotypes of mtDNA deletions are Kearns-Sayre Syndrome (KSS), Pearson syndrome, and progressive external ophthalmoplegia (PEO; *Marni, 2003*). Large stretches of deletions seen in different diseased patients suggest fragile sites in the mitochondrial genome similar to the fragile sites (FRA) sites of the nuclear DNA (*Richards and Macaulay, 2001*).

The FRA regions and other frequent chromosomal translocations have been observed in late replicating sites in the nuclear DNA, which are prone to replication fork stalling due to possible, stable secondary structure formation or inadvertent action of structure-specific nucleases (*Dillon et al., 2010*; *Lin et al., 2013*; *Raghavan et al., 2004*; *Sun et al., 2001*). Mitochondrial DNA has its replication machinery that is not governed by cell cycle checkpoints. Therefore, it is likely that the formation of a secondary structure may account for observed mitochondrial fragility.

The presence of alternate DNA structures like G-quadruplex, triplex, and cruciform structures in the mitochondrial genome has been reported along with their association with different mitochondrial diseases (*Bharti et al., 2014*; *Damas et al., 2012*; *Dong et al., 2014*; *Oliveira et al., 2013*). Breakpoint regions reported in the case of PEO patients were in the proximity of G-quadruplex motifs, further suggesting the role of non-B DNA structures in mitochondrial disorders (*Bharti et al., 2014*). G4 forming motifs were also located adjacent to mitochondrial deletion breakpoints associated with KSS, a clinical subgroup of mitochondrial encephalomyopathies (*Van Goethem et al., 2003*; *Zeviani et al., 1988*). Further, a G4 ligand RHPS4 preferentially localises to mitochondria and affects mitochondrial genome content, transcription elongation, and respiratory function (*Falabella et al., 2019a*; *Oliveira et al., 2013*). Moreover, in vitro results suggested that a single-base mutation in mtDNA (mt10251) was sufficient to form G4 DNA (*Chu et al., 2019*). Importantly, ATP-dependent G4 resolving helicase Pif1 could unfold non-B DNA structures and continue the synthesis by mitochondrial DNA polymerases, Pol γ and PrimPol (*Butler et al., 2020*). However, further studies are required to confirm the presence of these structures in vivo and establish their contribution towards the mechanism of breakage and deletions.

Mitochondrial DNA (mtDNA) 9 bp deletion caused by loss of one copy of the 9 bp repeat sequence (CCCCCTCTA) in the intergenic region of cytochrome c oxidase II (COII)/tRNA[Lys] is one of the most common deletions in mitochondria (*Redd et al., 1995*; *Thomas et al., 1998*). It has been associated with various diseases, including hepatocellular carcinoma (*Jin et al., 2012*; *Komandur et al., 2011*; *Krishnan and Turnbull, 2010*; *Zhuo et al., 2010*). The 9 bp deletion is also widely used as a phylogenetic marker to study evolutionary trends and migration of populations (*Soodyall et al., 1996*; *Yao et al., 2000*). The occurrence of deletions in such high frequency indicates the fragility of the region; however, its mechanism is unclear.

Out of the large-scale mtDNA deletions reported, one of the well-studied examples is mt[4977], and accumulation of it results in different mitochondrial disorders, including cancer (*Yusoff et al., 2019*). mt[4977] eliminates mitochondrial sequences between 8470 and 13,447 bp, which include the removal of genes required for normal oxidative phosphorylation (OXPHOS) (*Lee et al., 1994*; *Yang et al., 1994*; *Yen et al., 1991*). Recently the role of replication slippage and DSB repair has been shown to play a role in the generation of mtDNA[4977] in mammalian cells (*Phillips et al., 2017*). Previous studies have shown that the presence of a 13 bp direct repeat plays a major role in mt[4977], as only one of the repeats was retained following joining, indicating the involvement of microhomology-mediated end joining (*Samuels et al., 2004*; *Srivastava and Moraes, 2005*). When restriction endonucleases were used to introduce DSBs, there was the accumulation of deletions, which further implicates the imperfect repair and deletion in the mitochondrial genome (*Srivastava and Moraes, 2005*). A previous study from our laboratory suggested the existence of microhomology-mediated end joining (MMEJ) in mitochondria, which operate in a microhomology-dependent manner, explaining the frequently seen deletions in mitochondrial DNA (*Tadi et al., 2016*).

Here, we investigate the mechanism of fragility associated with the most common deletion seen in the mitochondrial genome. The putative role of non-B DNA structures and the mechanism of DNA breakage associated with the formation of mtDNA deletions are extensively studied. We report that the region associated with 9 bp deletion can fold into a G-quadruplex DNA structure using various biochemical and ex vivo methods. For the first time, we show that Endonuclease G can specifically bind to the G-quadruplex structure and cleave the mtDNA. Finally, the joining of the broken DNA could lead to the formation of '9 bp deletion' seen in human mitochondria.

## Results

Previously, different non-B DNA motifs, including G4 DNA motifs have been reported in the mitochondrial genome (*Dahal et al., 2022*; *Damas et al., 2012*; *Dong et al., 2014*). It was also suggested that alternate forms of DNA may play a role in the generation of deletions in mitochondria (*Dahal et al., 2022*; *Damas et al., 2012*; *Dong et al., 2014*). In silico studies along with biochemical analyses revealed that 5 classical G4 DNA motifs are present in the mitochondrial genome, which could fold into G4 DNA (*Cer et al., 2013*; *Dahal et al., 2022*; *Figure 1—figure supplement 1A*). Since '9 bp deletion' (8271–8281) is the most frequent mitochondrial deletion and occurs next to one of the 5 G4 motifs with the coordinate position 8252–8295 (designated as mitochondrial Region I) (*Figure 1A and B*; *Dahal et al., 2022*), we have investigated the putative role of G4 DNA structure in DNA breakage and formation of the mtDNA deletions. Notably, the chosen G-quadruplex motif is also present in two other large-scale deletions associated with mitochondria.

### G-quadruplex motifs from Region I of mitochondrial DNA, when present on shorter DNA, can fold into a G-quadruplex structure

To investigate whether mitochondrial Region I can indeed form G-quadruplex structures, oligomeric DNA containing the G-rich region referred to as 'G1' and complementary C strand (C1) were synthesised (*Figure 1B*). Radiolabelled oligomeric DNA was then subjected to electrophoretic mobility gel shift assay in the absence or presence of KCl. Faster mobility of the G-rich strand compared to its complementary C-rich strand suggested the formation of both intra and intermolecular G4 DNA structure in a KCl-dependent manner (*Figure 1C*). Importantly, the formation of intramolecular G4 DNA was abrogated when G-stretches were mutated (*Figure 1—figure supplement 1B, C*). Further, circular dichroism (CD) studies revealed that the spectra exhibited the characteristic of a parallel G-quadruplex DNA with absorption maxima of 260–270 nm and absorption minima of 240 nm as opposed to the C-rich strand, which showed the maxima similar to that of a typical single-stranded DNA (*Figure 1D*).

DMS protection assay was performed to understand the precise base pairing of guanine nucleotides involved in the G4 plates compared to the ones within connecting loops that facilitate DNA strand folding and, thus, G-quadruplex formation (*Nambiar et al., 2011*; *Nambiar et al., 2013*). DMS methylates the N7 position of the guanine base, making it susceptible to subsequent piperidine cleavage. Since the N7 position of G residues forming the G4 plate is engaged upon Hoogsteen base pairing, these residues are protected from DMS treatment and piperidine cleavage. When the G-quadruplex sequences from mitochondrial Region I (G1) were subjected to DMS protection assay, following structure formation, we observed that most G residues in this region showed a significant reduction in cleavage, suggesting protection of these nucleotides from DMS. Based on the observed protection of guanine from DMS, we conclude that these guanines are indeed involved in Hoogsteen bonding (*Figure 1E*, lane 2). The results suggest that the guanine residues from the G stretches of the mitochondrial Region I motif participate in G-quadruplex formation. Based on CD studies and gel shift assays, a parallel, intramolecular G-quadruplex structure was modelled (*Figure 1F*). Thus, our study reinforces that the Region I of mitochondrial DNA can fold into an intramolecular G-quadruplex (*Figure 1E and F*).

### The G4 motif present in Region I of mtDNA can fold into a G-quadruplex structure when present on a plasmid DNA

In the context of double-stranded DNA, G-quadruplexes can form only when the Hoogsteen base pairing between the guanine residues is stronger than the Watson and Crick H-bonding required for

**Figure 1.** Biochemical studies to investigate formation of G-quadruplex structure in Region I of the mitochondrial genome. (**A**) Sequence of the mitochondrial Region I with direct repeats and inverted repeats. Blue-shaded arrows represent the direct repeats, and yellow arrows represent the inverted repeats. The sequence in blue is G rich sequence predicted to form a G-quadruplex structure. The sequence after 9 bp deletion is also shown. (**B**) The oligomeric sequence of the G-rich strand (indicated by 'G1') and the complementary C-rich strand (indicated by 'C1'). These oligomers are derived from the predicted mitochondrial G-quadruplex forming Region I. The stretches of guanines are marked in blue. (**C**) The G and C-rich strands were incubated in the presence of 100 mM KCl and resolved on 15% native polyacrylamide gels in the absence (-KCl) or presence (+KCl) of KCl (100 mM) in the gel and running buffer. The substrate, intramolecular (Intra G), and intermolecular (Inter G) quadruplex structures are indicated. (**D**) CD spectra for G1 in the presence (red) or absence of 100 mM KCl (black). CD spectra for G1 (green) and C1 (black) in the presence of 100 mM KCl. In each case where KCl was added, the respective oligomer resuspended in Tris-EDTA buffer was incubated for 1 h at 37 °C and spectra were recorded using JASCO J-810 spectropolarimeter (scan range of 220–300 nm). (**E**) DMS protection assay for the Region I of mitochondria. The oligomer was allowed to fold into G4 DNA and then treated with DMS, followed by cleavage with piperidine. The products were resolved on a 15% denaturing PAGE. Substrate indicates the gel-purified DNA from the reaction incubated with or without KCl. All the positions of guanines are indicated. (**F**) A representative 2D model for intramolecular G-quadruplex structure formed at mitochondrial Region I based on the reactivity of guanine to DMS and CD studies for determining the orientation of DNA fold. Refer also *Figure 1—figure supplement 1*.

The online version of this article includes the following source data and figure supplement(s) for figure 1:

**Source data 1.** EMSA and CD studies to show formation of G-quadruplex structure in Region I.

**Source data 2.** Gel shift assay to show abrogation of G-quadruplex structure in mutants of Region I.

**Figure supplement 1.** Evaluation of G-quadruplex formation at mitochondrial Region I.



**Figure 2.** Evaluation of formation of G-quadruplex DNA in Region I of the mitochondrial DNA when cloned into a plasmid. (**A**) Schematic showing cloning of the mitochondrial Region I and its mutant to generate plasmids pDI1 and pDI2, respectively. The duplex region containing the G stretches is depicted in blue, while the mutated nucleotides are marked in red. (**B**) Schematic showing the primer extension across plasmid DNA containing mitochondrial Region I. Positions of primer, VKK11 is indicated. The radiolabeled primer (indicated with an asterisk) binds to one of the strands of the plasmid DNA upon heat denaturation and extends till it encounters a non-B DNA, as it blocks the progression of the polymerase. The products (dotted lines) are then resolved on a denaturing PAGE. (**C**) The plasmid, pDI1, containing the mitochondrial Region I, was used for primer extension studies using radiolabeled primers for the G-rich (VKK11) in a KCl (25, 50, 100 mM) dependent manner. A sequencing ladder was prepared using VKK11 by the chain termination method of sequencing. Pause sites are shown with dotted rectangles in turquoise. The sequence corresponding to the pause site is boxed, and sequence complementary to G-quadruplex forming motifs are indicated in pink. Sequencing ladder was used as marker. (**D**) Plasmids, pDI1 and its mutant pDI2, containing mutation in G4 motif, were used for primer extension studies using radiolabelled primer (VKK11). Pause sites are shown with dotted rectangles (turquoise). (**E**) Evaluation of the effect of different cations (Na+, K+, Li+) on non-B DNA formation at mitochondrial Region I when present on a plasmid (pDI1) by using primer VKK11 in a primer extension assay. The pause sites are indicated in dotted rectangle (turquoise). In panels C-F, 'M' denotes the 50 nt ladder. (**F**) Bar diagram showing the effect of ions in primer extension assay. The values are expressed in PSLU (photo-stimulated luminescence units) representing the extent of pause. Refer also *Figure 2—figure supplement 1*.

The online version of this article includes the following source data and figure supplement(s) for figure 2:

**Source data 1.** Primer extension studies to show formation of G-quadruplex structure in Region I containing plasmid.

**Figure supplement 1.** Schematic showing cloning of the mitochondrial Region I and its mutant to generate plasmids, pDI1 and pDI2.

duplex B-DNA formation. Therefore, the G-quadruplex forming ability of the mitochondrial Region I was tested after cloning into a plasmid, pDI1. Interestingly, an inverted repeat sequence (coordinate positions 8295–8363), which could fold into a cruciform DNA, was also present adjacent to the G4 motif. Thus, that was also part of pDI1 (*Figure 2—figure supplement 1*). Further, to delineate the role of G4 DNA in the plasmid context, a mutant plasmid with a scrambled sequence corresponding to the

G4 motif but with intact inverted repeats (pDI2) was also constructed (*Figure 2A*, *Figure 2—figure supplement 1*). Previous studies have shown that the formation of G4 DNA can lead to a pause during replication (*Kumari et al., 2015*; *Nambiar et al., 2013*; *Voineagu et al., 2008*; *Wells, 2009*). To assess whether the G4 DNA formed at Region I of the mitochondrial genome can act as a replication pause, a primer extension assay was performed using pDI1 (*Kumari et al., 2015*; *Figure 2B and C*). A sequencing ladder was also generated for mapping the location of the polymerase pause sites on the plasmid (*Figure 2C*). Results revealed multiple polymerase pause sites, and the intensity of the pauses increased with increasing concentrations of KCl (25, 50 and 100 mM; *Figure 2C*). Interestingly, there was significantly fewer sites and less pausing observed in the absence of KCl, indicating the absence of the formation of intramolecular G-quadruplex species in these cases (*Figure 2C*, lane 2). Considering that the formation of cruciform DNA occurs irrespective of the presence of KCl, these results also suggest that the observed pause was due to the formation of G4 DNA, but not cruciform DNA. Besides, the observed polymerase pauses mapped to the G-quadruplex motif (*Figure 2C*, lanes 7–9). Abrogation of G4 DNA structure by mutation of G4 DNA motif (pDI2) resulted in the disappearance of the observed pause site, further confirming the above results (*Figure 2D*, lanes 3, 4 and lane 6, 7). It was also observed that the appearance of strong polymerase pause sites were dependent on $K^+$ (*Figure 2E*, lanes 4–5; F), but not on other monovalent cations such as $Li^+$ (with no or little G-quadruplex stabilising properties) or $Na^+$ (with limited stabilising properties) (*Figure 2E*, lanes 6–9; F) suggesting that the non-B DNA responsible for polymerase arrest was G4 DNA.

## Mitochondrial Region I can exist as a G-quadruplex structure within the mitochondrial genome and, when present on a plasmid DNA

Formation of G4 DNA on a plasmid or mitochondrial genome could result in single-strandedness at complementary and adjacent regions corresponding to the G4 motif. Sodium bisulphite modification assay was performed on pDI1 to check the single-strandedness in the complementary C strand of Region I. Sodium bisulphite deaminates cytosine resulting in uracil, when present on single-stranded DNA (*Figure 3—figure supplement 1A*). This change can be read as a C→T conversion after PCR and sequencing (*Figure 3—figure supplement 1A, B*). Of the 40 clones sequenced, 38 molecules showed C to T conversion to varying extents in either top or bottom strands (*Figure 3A*; *Figure 3—figure supplement 1C*). Interestingly, the cytosines complementary to the G-quadruplex motif exhibited single-stranded nature (*Figure 3A*), suggesting the formation of a G-quadruplex structure in the complementary G-rich strand. Importantly, ~15% of molecules showed continuous conversion at the region corresponding to the G4 motif. These results suggest that Region I of mitochondrial DNA can fold into a non-B DNA even when cloned into a plasmid, consistent with the result obtained following primer extension.

Further, to investigate the existence of G4 DNA structures in the mitochondrial genome, a sodium bisulphite modification assay was performed after extracting the mitochondrial DNA from human cells (Nalm6) under non-denaturing conditions. Results showed that out of 59 DNA molecules analysed following cloning and sequencing, 52 showed C→T conversions to varying extents (*Figure 3B and C*; *Figure 3—figure supplement 1C*). Cumulative analysis of converted cytosines on a fragment of 281 bp revealed single-strandedness at Region I corresponding to the complementary region of G4 motifs. When the mitochondrial genome was analysed, regions adjacent to the G4 motif also showed certain levels of single-strandedness (*Figure 3B*). However, the C→T conversion rate was significantly lower when upstream and downstream sequences were analysed for conversion. Further, among the 50 clones from the top strand, 14 (28%) had continuous conversions of 11 or more cytosines when a stretch of 20 cytosines modifications was considered on a single DNA molecule level (*Figure 3C*; *Figure 3—figure supplement 1C*). This suggests that the formation of G4 DNA at Region I may not occur in all DNA molecules of the mitochondrial genome at a given time. However, based on these results in conjunction with biochemical assays, it is evident that the G-quadruplex motif at Region I can indeed fold into a G-quadruplex structure in mitochondria.

## BG4, a G-quadruplex binding antibody, binds to G-quadruplex structures in the mitochondrial genome

The previous studies established that BG4 antibody can bind to G-quadruplexes formed in DNA and RNA (*Chambers et al., 2015*; *Das et al., 2016*; *Javadekar et al., 2020*). To study whether BG4 indeed



**Figure 3.** Evaluation of G-quadruplex formation at mitochondrial Region I. (**A**) Bisulphite modification assay on plasmid (pDI1) containing mitochondrial Region I. Vertical bar represents the number of times the respective cytosine in the top and bottom strands of mitochondrial Region I is converted to thymine after deamination when treated with sodium bisulphite, followed by PCR. (**B, C**) Bisulphite sequencing on the mitochondrial genome for determining the formation of G4 DNA at Region I. Each vertical bar represents the conversion of a cytosine to uracil in the top strand or bottom strand. A total of 59 clones were sequenced from both the top and bottom strands (**B**). Single strandedness was observed in a DNA fragment of 198 nt containing Region I of the mitochondrial genome following bisulphite modification assay (**C**). Each row represents cytosines present in a DNA molecule. Each dark circle represents the conversion of cytosine to thymine on the top strand after deamination upon treatment with sodium bisulphite, followed by PCR and DNA sequencing. Of the 59 clones sequenced, the most reactive 25 molecules are shown (**C**). In all the panels, sequences corresponding to the G-quadruplex forming motif are indicated in a dotted rectangular box (mustard yellow). Refer also *Figure 3—figure supplement 1*.

The online version of this article includes the following source data and figure supplement(s) for figure 3:

**Source data 1.** Sequence of clones after bisulphite treatment in a plasmid containing Region I of mitochondria.

**Source data 2.** Sequence of clones after bisulphite treatment in a region I of mitochondria.

**Figure supplement 1.** Schematic showing conversion of cytosine to thymine following bisulphite modification assay.

binds to the mitochondrial genome inside cells, immunofluorescence in HeLa and HEK 293T cells was performed after testing the specificity of the antibody (*Javadekar et al., 2020*). Mito-tracker Deep Red or Mito-tracker Green FM was used for staining mitochondria, while the nucleus was stained with DAPI. BG4 was stained with Alexa fluor-tagged secondary antibodies. Merging of green and red foci



**Figure 4.** Evaluation of G-quadruplex structure in mitochondria of cells. (**A**) Representative images of HeLa and HEK293T cells showing colocalisation of BG4, the G4 binding antibody to mitochondria following immunofluorescence assay. The nucleus is stained with DAPI (blue colour), mitochondria with MitoTracker DR (red) and BG4 with Alexa-Fluor 488 (green). A merged image is shown with a merge of red and green, as depicted by Coste's mask (colocalisation is represented as a white dot). (**B, C**) Quantitation showing colocalisation of BG4 with MitoTracker indicated as dot plots. The colocalisation was quantified using Mander's colocalisation coefficient (ImageJ software) analysing a minimum of 100 cells as red over green (**B**) and green over red (**C**). (**D**) Representative image of Rho(0) cells showing localization of BG4 to mitochondria following immunofluorescence as investigated in panel (**A**). (**E**) Quantitation showing comparison of colocalization of BG4 between HeLa cells and Rho(0) cells shown as dot plots. The colocalization was quantified using Mander's colocalization coefficient analyzing a minimum of 50 cells as red over green. Refer also *Figure 4—figure supplement 1*.

The online version of this article includes the following source data and figure supplement(s) for figure 4:

**Source data 1.** Localization of BG4 to mitochondria.

**Source data 2.** Immunofluorescence showing the localization of BG4 to mitochondria.

**Figure supplement 1.** Evaluation of existence of G-quadruplex in mitochondria of cells.

resulting in yellow foci was observed in the case of both HeLa and HEK 293T cells, which revealed localisation of BG4 to mitochondria (*Figure 4A*, *Figure 4—figure supplement 1A*). The colocalisation was further analysed using the JaCoP plugin in ImageJ software. In both cases, an image with Coste's mask is presented, showing the extent of localisation in the form of a white dot after merging red and green foci (*Figure 4A–C*). Colocalisation analyses show Mander's coefficient value between 0.25–0.4, irrespective of the stain used as MitoTracker, in the case of HeLa and 293T cells indicating that G-quadruplex structures can be detected in the mitochondrial genome in cellular context (*Figure 4B and C*). To further confirm that the colocalisation was indeed between the MitoTracker and BG4 antibody, immunofluorescence was performed in Rho(0) cells (human osteosarcoma cells with depleted mtDNA)



**Figure 5.** Evaluation of BG4 binding to G-quadruplex structure in mitochondrial genome. (**A**) Schematic showing the experimental strategy used for mito-IP using anti-BG4. Briefly, cells were crosslinked and then mitochondria were isolated and sonicated to obtain the small fragments of mitochondrial DNA. Purified BG4 antibody was used along with protein A/G agarose beads to pull down the BG4 bound regions. (**B**) BG4 bound mtDNA was purified after reverse crosslinking and used for real-time PCR using primers derived from different regions of the mitochondrial genome, which include 5 G-quadruplex forming regions and 10 random regions. Input DNA served as template control. No antibody control was also used. Bars in blue (first 5) are for G-quadruplex forming regions, while green (last 10) are for random regions. Y-axis depicts threshold Ct value obtained following real time PCR for each primer. Error bar represents mean ± SEM. (**C**) Agarose gel profile showing the amplification of Input DNA (left panel) and BG4 pull down DNA (right panel). 'M' denotes 100 bp ladder. Refer also **Figure 5—figure supplement 1**.

The online version of this article includes the following source data and figure supplement(s) for figure 5:

**Source data 1.** BG4 ChIP to show the binding of BG4 to mitochondrial G-quadruplex forming regions.

**Figure supplement 1.** Evaluation of existence of G-quadruplex in mitochondrial DNA.

(**Dong et al., 2017**; **Tan et al., 2015**). Results suggested that although BG4 foci were observed in the cytoplasm of Rho(0) cells, they do not seem to colocalize with mitochondria (**Figure 4D and E**).

To examine whether BG4 can indeed bind to mitochondrial G-quadruplexes, in vitro binding of purified BG4 to oligomers harbouring the G-quadruplex region was performed (**Figure 4—figure supplement 1B**). Results indicated that the bound complex was seen when G-rich oligomer (G1) was incubated with BG4, while such binding was absent when incubated with C-rich oligomer (C1) (**Figure 4—figure supplement 1B**).

To investigate whether BG4 can bind to G-quadruplex DNA present in the mitochondrial genome, the mito IP-qPCR strategy was employed (**Figure 5A**, **Figure 5—figure supplement 1**). Briefly, following crosslinking, mitochondria were isolated, and the region binding to BG4 was immunoprecipitated. Five regions corresponding to G4 motifs and ten random regions without G4 motifs were selected for analysis by using appropriate PCR primers (**Figure 5—figure supplement 1**). Results showed that all five G-quadruplex forming regions amplified between 10–18 cycles when qPCR was performed (**Figure 5B**). In contrast, 9 of 10 random regions did not show any amplification following

the BG4 pull-down (*Figure 5B*). Sequence analysis showed that the control region (CR7) amplified after the BG4 pull-down had high GC content. Besides, when CR7 was analysed for potential non-B DNA formation, we noted the presence of two-plate G4 DNA motif. CR7 may likely fold into a 2-plate G4 DNA, which could explain its binding with BG4 (*Figure 5B*). However, further studies are required to confirm this hypothesis. Samples with input DNA served as a positive control, and as expected, all the regions were amplified. Samples where no antibody was added, did not result in amplification when G4-specific or random region primers were used, revealing the specificity of BG4 towards G4 DNA (*Figure 5B*). Similar results were also obtained when BG4 pull-down samples were evaluated using semiquantitative PCR (*Figure 5C*). However, none of the regions amplified when samples from secondary control were used, including CR7 (*Figure 5C*). Sequence analysis showed that the control region amplified following BG4 pull down had high GC content and predicted for two plates G4 structure, which requires further analysis (*Figure 5B and C*).

Thus, our results suggest that Region I and other G4 motifs in the human mitochondrial genome could fold into G-quadruplex DNA. However, based on bisulphite DNA sequencing from Region I, the structure may not be formed in all DNA molecules, and the frequency may be <25%.

## Mitochondrial extract induces cleavage at or near the G-quadruplex structure at Region I

To elucidate the mechanism responsible for cleavage at mitochondrial Region I, the involvement of mitochondrial nucleases was considered. To test this, the mitochondrial extract was isolated from the rat tissues (testes and spleen) and used for cleavage assay. The extract was confirmed to be mitochondrial in nature and was tested for cross-contamination by western blotting using the anti-cytochrome C (mitochondrial marker), anti-PCNA (nuclear or cytosolic marker), and anti-Actin (loading control) (*Figure 6A*). Plasmid harbouring mitochondrial Region I (pDI1) was treated with the mitochondrial extract (0.25, 0.5, 1, and 2 µg) and resolved on an agarose gel following deproteinisation. Interestingly, a concentration-dependent decrease in supercoiled form and increase in the nicked form of plasmid DNA was observed when treated with mitochondrial extracts (*Figure 6B*, lanes 1–5). Further, there was also a concentration-dependent increase in the linear form of the plasmid DNA, indicating presence of two different nicks at proximity leading to a DSB (*Figure 6B*, lanes 1–5, C). Although there is a possibility that more than one nick could be present in the pDNA, since the 6 bp mutation in the G4 motif (*Figure 6—figure supplement 1A*) resulted in abrogation of the conversion of supercoiled to nick DNA, we assume that there will be only a single nick present per strand of the plasmid DNA. To test whether the cleavage observed was due to the presence of G4 DNA, the plasmid bearing mutation in the G4 motif, but bearing intact adjacent inverted repeats (pDI2), was treated with mitochondrial extract and analysed. Interestingly, there was a significant reduction in cleavage when two stretches of G's were mutated (*Figure 6B*, lanes 6–10, 6 C; *Figure 6—figure supplement 1A*), indicating the cleavage was indeed due to the presence of G4 DNA.

For deciphering whether the position of cleavage observed was indeed at Region I, primer extension assay was carried out on pDI1 and its mutant (pDI2 and pDR4; *Figure 6—figure supplement 1A*) following cleavage by mitochondrial extracts isolated from testes and spleen. Results revealed two specific nicking sites when incubated with mitochondrial extracts from both testes and spleen. These were mapped to the position corresponding to the G-quadruplex forming region when the DNA sequencing ladder was used as a marker (*Figure 6D*). Interestingly, mutation of the G4 motif resulted in a significant reduction of cleavage activity, as one of the two cleavage sites disappeared when 2 G stretches were mutated (pDI2) from the G4 motif (*Figure 6E and F*; *Figure 6—figure supplement 1*). However, the upper cleavage site remains unchanged with the mutation. This may be due to G4 structure formation in alternate conformations, as Region I can fold into multiple forms (*Figure 6—figure supplement 2*; *Dahal et al., 2022*). Interestingly, generation of mutation in third stretch of guanines in addition to the above-described mutation (pDR4) resulted in remarkable reduction in the cleavage activity and led to disappearance of both the bands (*Figure 6—figure supplement 1*). These results confirm the involvement of G4 DNA in the observed cleavage at Region I.

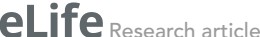

**Figure 6.** Evaluation of mitochondrial extract induced cleavage at G4 DNA formed at Region I of the mitochondrial genome. (**A**) Purification and characterization of mitochondrial extracts. Mitochondrial extract was prepared from rat testis and spleen, and its purity was evaluated using specific markers by western blotting. Purity of CE (cytosolic extract) and ME (mitochondrial extract) were determined using antibodies against PCNA (nuclear and cytoplasmic marker), cytochrome C (mitochondrial marker) and Actin (loading control). (**B**) In vitro nicking assay on a plasmid containing wild type (WT pDNA) mitochondrial Region I (pDI1) and mutant (Mt pDNA) of Region I (pDI2). Both wild type and mutant plasmids were incubated with mitochondrial extract from rat testes (ME) and resolved on a 0.8% agarose gel. Lanes 1 and 6 served as the control without any extract in the reaction, whereas lanes 2–5 and lanes 7–10 are with increasing concentration (0.25, 0.5, 1, and 2 µg) of mitochondrial extract for pDI1 and pDI2, respectively. 'OC' is open circular, 'LIN' is linear, and 'SC' is supercoiled. (**C**) Quantification showing the efficiency of mitochondrial extracts (Mt extracts) mediated cleavage when a plasmid containing mutant and wild type G4 motif derived from Region I was compared. The values in Y-axis are expressed in PSLU (photo-stimulated luminescence units) representing the extent of pause. (**D**) Primer extension assay on plasmid containing mitochondrial Region I (pDI1) to determine the position and location of cleavage. pDI1 was incubated with either mitochondrial extract prepared from testes or spleen in an increasing concentration (0.25, 0.5, 1, and 2 µg), reaction products were purified and used for primer extension assay. Pause sites are indicated, which correspond to G4 DNA motif. Sequencing ladder was used as marker. Complementary sequence corresponding to G-quadruplex forming motif is indicated in blue. 'M' is 50 nt ladder. (**E**) Primer extension assay on pDI1 and pDI2 (G4 mutant) following incubation with mitochondrial extract prepared from rat testis (1 µg). 'M' is 50 nt ladder. '+' indicates addition of 1 µg mitochondrial extract shown in triplicates. (**F**) Quantification showing the cleavage efficiency of mitochondrial extracts (testes) when a plasmid containing mutant and wild type G4 motif derived from Region I was incubated. Cleavage intensity is shown in Y-axis. Error bar represents ± SEM. Refer also *Figure 6—figure supplements 1 and 2*.

The online version of this article includes the following source data and figure supplement(s) for figure 6:

**Source data 1.** Gel profiles showing the mitochondrial induced cleavage at mitochondrial region I.

**Source data 2.** Cleavage assay on plasmid bearing wildtype and mutant G4 sequence.

*Figure 6 continued on next page*

*Figure 6 continued*

**Figure supplement 1.** Mitochondrial extract mediated cleavage assay on plasmids bearing different mutations at G4 DNA motif containing plasmid.

**Figure supplement 2.** DNA sequences that support formation of different conformation of G-quadruplex DNA structures in Region I.

## Endonuclease G mediates structure-selective cleavage at mitochondrial Region I

Among different mitochondrial endonucleases, Endonuclease G was shown to bind preferentially to the guanine-rich strands by a previous study (*Ruiz-Carrillo and Renaud, 1987*). Since we have observed the formation of G-quadruplex DNA structures with multiple conformations (*Figure 6—figure supplement 2*; *Dahal et al., 2022*) at Region I of the mitochondrial genome, we considered the possibility of Endonuclease G as the protein responsible for the observed cleavage. In a different study, specific cleavage at kinked DNA by Endonuclease G has also been reported (*Ohsato et al., 2002*). Based on these observations, we wondered whether Endonuclease G could cleave G4 DNA present at Region I on pDI1. For this, Endonuclease G and its catalytic dead mutant (ΔH151A) were overexpressed in bacteria, purified, and identity was confirmed by western blotting (*Figure 7—figure supplement 1A–C*). Purified Endonuclease G was subjected to a plasmid nicking assay and analysed on an agarose gel. Results showed that while the wild-type plasmid (pDI1) showed a conformation change from supercoiled to open circular, mutant plasmid (pDI2) retained its supercoiled form (*Figure 7A and B*). Besides, the linear form of the plasmid DNA was also observed when a higher concentration (60, 90, 120 ng) of Endonuclease G was used for the cleavage in the case of pDI1 (*Figure 7A*).

Interestingly, primer extension studies revealed that Endonuclease G cleaved mitochondrial DNA at multiple sites corresponding to a region of G-quadruplex DNA formation at Region I when it was present on pDI1 (*Figure 7C*, lanes 1–3, 5–8). Importantly, the primary two cleavage positions were comparable to the one observed when the mitochondrial extract was used for the study (*Figure 7C*, lane 2–4). However, unlike mitochondrial extract, incubation with purified Endonuclease G resulted in cleavage at multiple positions across the G4 motif. It may be that multiple molecules of purified Endonuclease G may bind to different forms of G4 DNA resulting in cleavage at different positions. In the case of mitochondrial extract, additional proteins may bind to G4 DNA, thus preventing cleavage at multiple sites. More importantly, when a protein with a mutation at the nuclease domain of Endonuclease G (ΔH151A) was used, the cleavage activity was abrogated (*Figure 7D and E*; *Figure 7—figure supplement 1B, C*). Further, when endonucleases such as CtIP, FEN1 and RAGs were used for cleavage assay, results showed that, unlike Endonuclease G, none of the purified endonucleases exhibited any significant cleavage (*Figure 7—figure supplement 1J*), although the presence of CtIP (*Tadi et al., 2016*) and FEN1 (*Kalifa et al., 2009*; *Klungland and Lindahl, 1997*; *Tadi et al., 2016*) has been shown previously in mitochondria. In all cases, the identity and activity of the endonucleases used were confirmed before use (*Figure 7—figure supplement 1D–I*).

Immunodepletion of Endonuclease G from mitochondrial extracts prepared from rat testes showed a significant reduction in cleavage efficiency at the mitochondrial G-quadruplex Region I, compared to beads alone control (*Figure 7F–I*). However, immunodepletion of MGME1, another important nuclease known to be present in mitochondria, did not result in any significant difference in the cleavage efficacy (*Figure 7G–I*; *Phillips et al., 2017*; *Yang et al., 2018*). Importantly, the activity was restored when purified Endonuclease G was added back following its immunodepletion from the extract, confirming the role of Endonuclease G in the cleavage (*Figure 7J and K*).

## Endonuclease G is expressed in mitochondria and colocalises with mitochondrial matrix protein TFAM within the cells

Considering that Endonuclease G is one of the essential mitochondrial nucleases (*McDermott-Roe et al., 2011*; *Misic et al., 2016*; *Wiehe et al., 2018*; *Zhou et al., 2016*), the presence of Endonuclease G was examined in mitochondria, using mitochondrial specific marker, 'MitoTracker Deep Red or MitoTracker green FM' and using immunofluorescence with Endonuclease G antibody in three cell lines of different origin (*Figure 8A*; *Figure 8—figure supplement 1*). Merge of MitoTracker with tagged Endonuclease G revealed localization of Endonuclease G in mitochondria in HeLa (human cervical cancer), HEK 293T (human embryonic kidney epithelial) and MEF (mouse embryonic fibroblast) cell lines (*Figure 8A and B*; *Figure 8—figure supplement 1*). Further, to check if Endonuclease



**Figure 7.** Studies to identify mitochondrial nuclease responsible for cleavage at G4 DNA formed at Region I of the mitochondrial genome. (**A**) In vitro nicking assay using purified Endonuclease G on wild type and mutant plasmids containing mitochondrial Region I (pDI1 and pDI2). Both wild type and mutant plasmids were treated with increasing concentration of purified Endonuclease G and resolved on a 0.8% agarose gel. Lanes 1 and 6 served as the control without any protein in the reaction whereas lanes 2–5 and lanes 7–10 are with increasing concentration (30, 60, 90, 120 ng) of purified Endonuclease G. (**B**) Quantification showing the efficiency of Endonuclease-G-mediated cleavage when a plasmid containing mutant and wild type G4 motif derived from Region I was compared. (**C**) Cleavage assay was performed on a plasmid containing mitochondrial Region I, pDI1 following incubation with purified Endonuclease G. Primer extension assay was carried out using [γ-$^{32}$P] radiolabeled VKK11 primer and resolved on 8% denaturing polyacrylamide gel. Lanes 2 and 3 represent 30 and 60 ng of Endonuclease G incubated samples. Lane 1 is without protein, lane 4 is with mitochondrial extract (ME). and lanes 5–8 are A, C, G, T represents the sequencing ladder. 'M' is 50 bp ladder. Marked regions represent the specific cleavage products. (**D**) Cleavage assay was performed on a plasmid containing mitochondrial Region I, pDI1 following incubation either with purified Endonuclease G or mutant Endonuclease G (ΔH151A). Primer extension assay was carried out using γ-$^{32}$P radiolabeled VKK11 primer and resolved on 8% denaturing polyacrylamide gel. Lane 2 is primer alone, Lanes 3 and 6 are without protein, lanes 4 and 5 are with 30 and 60 ng of Endonuclease G, lanes 7 and 8 are with 30 and 60 ng of mutant Endonuclease G incubated samples. 'M' is 50 bp ladder. (**E**) Quantification showing the efficiency of wild type and mutant Endonuclease G (ΔH151A) mediated cleavage when a plasmid containing wild type G4 motif derived from Region I was compared. (**F**) Western blotting showing immunodepletion of Endonuclease G from rat testicular mitochondrial extracts. Protein A/G beads were incubated with anti-Endonuclease G and then with the extracts. Actin served as a loading control. (**G**) Immunodepletion of another endonuclease present in mitochondria, MGME1 from rat testicular mitochondrial extracts as described in panel F. (**H**) Endonuclease G or MGME1-depleted extract was incubated with pDI1 and used for the primer extension using VKK11 primer. Extract without the addition of antibody served as bead control (lane 4). Lane 3 is no protein control, lanes 5 and 6 corresponds to increasing concentrations of MGME1 depleted extracts, while in lanes 7 and 8, increasing

*Figure 7 continued on next page*

Figure 7 continued

concentrations of Endonuclease G depleted extracts were added. 'M' is 50 nt ladder. Cleavage positions are indicated by arrow and boxed (red). (**I**) Bar diagram depicting quantitation showing the impact of immunoprecipitation of Endonuclease G based on multiple experiments. (**J**) Reconstitution assay was performed by the addition of purified Endonuclease G following its immunodepletion. Lane 2 represents the primer alone; Lane 3 represents the beads control; lane 4 represents the Endonuclease G depleted extract while lanes 5 and 6 represent the addition of purified Endonuclease G to the depleted extracts (performed in duplicate reaction). (**K**) Bar diagram representing the cleavage intensity after reconstitution assay as shown in panel J. In panels, C, D, H and J, 'M' represents 50 nt ladder. In panels E, I and K, quantitation is based on three biological repeats and data is shown with error bar calculated as mean ± SEM (*ns* not significant, *p<0.05, **p<0.005, ***p<0.0001). Refer also *Figure 7—figure supplement 1*.

The online version of this article includes the following source data and figure supplement(s) for figure 7:

**Source data 1.** Gel profiles showing the Endonuclease-G-induced cleavage at mitochondrial region I.

**Source data 2.** Gel profiles showing the purification of different endonucleases.

**Figure supplement 1.** Overexpression, purification and activity assay of different endonucleases.

G is present in the mitochondrial matrix, colocalisation analysis was performed along with TFAM, a transcription factor known to be present in the mitochondrial matrix by conducting immunofluorescence studies in HeLa cells (*Figure 8C*). Results revealed low but distinct levels of colocalisation of Endonuclease G and TFAM, indicating the presence of Endonuclease G in the mitochondrial matrix (*Figure 8C and D*). Western blotting further confirmed this after fractionation of the mitochondria (see below).

Further, shRNA-mediated knockdown of Endonuclease G was performed in HeLa cells. Knockdown efficiency was assessed by western blotting using Actin as the loading control (*Figure 8E*). HeLa cells transfected with scrambled plasmid served as transfection control. Cleavage assay using mitochondrial extracts following knockdown of Endonuclease G revealed a significant reduction in cleavage efficiency at Region I, as that of scrambled control, suggesting its role in mitochondria in the context of cells (*Figure 8F and G*).

## Endonuclease G binds to the G-quadruplex forming regions of mitochondria inside cells

To investigate if Endonuclease G binds to G-quadruplex forming regions of mitochondria, G-quadruplex specific antibody, BG4 was used for colocalisation studies along with anti-Endonuclease G (*Figure 9A and B*; *Figure 9—figure supplement 1A*). Yellow foci in the merged image suggested colocalisation of BG4 to Endonuclease G. Approximately 100 cells were analysed for colocalisation of anti-BG4 and anti-Endonuclease G using the Mander's colocalisation coefficient, for red over green, was ~0.32, while green over red was ~0.21 indicating the binding of Endonuclease G to G-quadruplex DNA structure (*Figure 9B*).

An antibody binding assay was performed to examine the binding of Endonuclease G to mitochondrial G quadruplex regions. Purified Endonuclease G was allowed to bind to mitochondrial DNA isolated from Nalm6 cells (*Figure 9—figure supplement 1B*) and cross-linked. Sonicated mitochondrial DNA was incubated with anti-Endonuclease G and Protein A/G beads. The resulting bound DNA was reverse crosslinked, purified and used for both real-time and semiquantitative PCR (*Figure 9—figure supplement 1C, D*). The threshold values presented as a bar diagram suggested that Endonuclease G can bind to all G quadruplex regions (GR1- GR5), including the one present at mitochondrial Region I, while the control regions (CR1-CR10) (not known to form canonical G quadruplex) did not bind to Endonuclease G (*Figure 9—figure supplement 1C–E*). These results reveal the specific binding of Endonuclease G to mitochondrial G quadruplex regions.

Further, the mitochondrial extract was incubated with the mitochondrial DNA, and a pull-down assay was done using Endonuclease G antibody (*Figure 9C*). Following the pull-down, beads were separated and loaded for protein analysis and western to confirm the pull-down (*Figure 9—figure supplement 2A*). The remaining was used for DNA isolation, followed by real-time and semiquantitative PCR. Results suggested the binding of Endonuclease G to G-quadruplex regions (*Figure 9C–E*). Endonuclease G binding assay inside cells was performed to assess the binding of Endonuclease G to mitochondrial G4 DNA (*Figure 9F*, *Figure 9—figure supplement 1C*). The antibody-bound mitochondrial DNA was used for real-time PCR to amplify the G quadruplex regions and control regions.



**Figure 8.** Evaluation of expression of Endonuclease G within different mammalian cells and Endonuclease-G-mediated cleavage at mtDNA following shRNA mediated knockdown within cells. (**A**) Localization of Endonuclease G in mitochondria in different cell lines. Representative images of localization of Endonuclease G to mitochondria in HeLa, MEF and HEK293T cells. FITC-conjugated secondary antibodies were used for detecting Endonuclease G proteins. MtDR is Mitotracker Red, a mitochondrial marker. DAPI is used as nuclear stain. (**B**) Colocalization analyses of Endonuclease G and Mitotracker signals using JaCoP in ImageJ software based on immunofluorescence studies performed in multiple cell lines (see **Figure 8—figure supplement 1**). Minimum of 50 cells were used for analysis of colocalization of red and green signals and plotted as the colocalization value of green overlapping red. Y-axis depicts the Mander's colocalization coefficient value calculated for green over red and plotted in the form of dot plot. The significance was calculated using GraphPad Prism 5.0 with respect to secondary control and shown as mean ± SEM (*ns* not significant, \*p<0.05, \*\*p<0.005, \*\*\*p<0.0001). (**C**) Representative images of colocalization of Endonuclease G to mitochondrial matrix (TFAM) in HeLa cells. Conjugated secondary antibodies were used for detecting Endonuclease G (Alexa Fluor 488) and mitochondrial matrix protein, TFAM (Alexa Fluor 568). DAPI is used as nuclear stain. (**D**) Colocalization analyses of Endonuclease G and TFAM signals using JaCoP in ImageJ software. Minimum of 50 cells were used for analysis of colocalization of red and green signals and plotted. Y-axis depicts the Mander's colocalization coefficient value calculated for green over red and plotted in the form of dot plot. Control represents the panel where only one of the primary antibodies was used. (**E**) Knockdown of Endonuclease G from HeLa cell using PEI mediated transfection. shRNA against Endonuclease G cloned plasmid was used for transfection. Cells were harvested after 48 h and mitochondrial extracts were prepared. Western blotting was performed to confirm the knockdown of Endonuclease G from the HeLa cells. Actin served as a loading control. (**F**) The knockdown extract was incubated with the plasmid and used for the primer extension using VKK11 primer ('I' and 'II' are two biological repeats). Extract prepared from the sample transfected with scrambled plasmid served as a control (SCR control). Lanes 3, 4, 7, and 8 serve as scrambled controls while lanes 5, 6, 9, and 10 are for knockdown extracts. I and II represent two independent batches of experiments. 'M' is a 50 nt ladder. (**G**) Bar diagram representing the cleavage intensity of the extracts prepared after transfection with scrambled plasmid and shEndo G plasmid. In panels F and G, a minimum of three biological repeats were performed and the data is shown with the error bar calculated as SEM (ns: not significant, \*p<0.05, \*\*p<0.005, \*\*\*p<0.0001). Refer also **Figure 8—figure supplement 1**.

*Figure 8 continued on next page*

*Figure 8 continued*

The online version of this article includes the following source data and figure supplement(s) for figure 8:

**Source data 1.** Localization of Endonuclease G to mitochondria.

**Source data 2.** Immunofluorescence showing the Localization of Endonuclease G to mitochondria.

**Figure supplement 1.** Analysis of Endonuclease G localization in mitochondria.

Results showed specific amplification of G quadruplex regions and no amplification of control regions, suggesting the specificity of Endonuclease G to G quadruplex regions within the cells (***Figure 9F***).

P1 nuclease assay was also performed to determine the binding footprint of Endonuclease G to G-quadruplex forming Region I by incubating the wild type and the mutant oligomers of Region I in the presence and absence of Endonuclease G. Interestingly, binding of Endonuclease G to G-rich strand (G1) partially rescued the sensitivity of P1 nuclease (***Figure 9—figure supplement 2B***, lanes 5–8). In contrast, in the case of C-rich strands, mutants and random oligomers, such protection was absent (***Figure 9—figure supplement 2B***).

Although Endonuclease G was thought to be localised within the mitochondrial intermembrane space, its proximity to the mitochondrial matrix was supported by multiple experimental evidence suggesting a direct interaction of Endonuclease G with mtDNA (***Duguay and Smiley, 2013***; ***McDermott-Roe et al., 2011***; ***Wiehe et al., 2018***). Therefore, as a preliminary investigation, we were interested in testing the conditions in which Endonuclease G could be released to the mitochondrial soluble fraction from the inner mitochondrial membrane. To do this, HeLa cells were exposed to Menadione, a mitochondria-specific ROS inducer (25 μM, for 2 hr). Cells were harvested, and fractionation of mitochondria was performed. Fractionated extracts were equalized and used for western blotting studies (***Figure 10A***). Results showed that exposure to stress conditions resulted in elevated levels of Endonuclease G in the matrix (***Figure 10A and B***). However, this warrants further detailed investigation. Thus, we hypothesize that different stress conditions may regulate the release of Endonuclease G to mitochondrial matrix.

Based on the present and previous studies (***García-Lepe and Bermúdez-Cruz, 2019***; ***Tadi et al., 2016***; ***Wisnovsky et al., 2018***), we propose that under stress conditions, Endonuclease G may be released to the matrix from the inner membrane space where it binds to mitochondrial DNA. This could help the induce DNA breaks by Endonuclease G at Region I of the mitochondrial genome. DNA modifying enzymes such as CtIP, MRN, and Exonuclease G may help in generating 3' overhangs, following which ligation of broken ends by an unknown mechanism leads to the '9 bp deletion' at the Region I of the mitochondrial genome (***Figure 11***).

## Discussion

The present study in conjunction with previous studies, suggest the existence of five G-quadruplex DNA structures in the mitochondrial genome (***Dahal et al., 2022***; ***Damas et al., 2012***; ***Dong et al., 2014***), all with three-plate conformation that follows the general empirical formula for G-quadruplexes $d(G_{3+}N_{1-7}G_{3+}N_{1-7}G_{3+}N_{1-7}G_{3+})$. G-quadruplexes with three-plate are considered energetically stable (***Puig Lombardi and Londoño-Vallejo, 2020***). A close analysis of patient breakpoint regions revealed the presence of high-frequency mitochondrial deletion junctions (9 bp deletion) adjacent to one of the G-quadruplex structures at Region I.

The presence of G-quadruplex motifs proximal to mitochondrial deletions has been reported (***Falabella et al., 2019a***; ***Falabella et al., 2019b***; ***Oliveira et al., 2013***). Computational analysis (***Bharti et al., 2014***) revealed that G4 sequences proximal to mtDNA deletions in several genetic diseases like Kearns-Sayre syndrome, Pearson marrow-pancreas syndrome, Mitochondrial myopathy, Progressive external ophthalmoplegia, etc. G4 motifs were also located adjacent to mitochondrial deletion breakpoints associated with KSS, a clinical subgroup of mitochondrial encephalomyopathies associated with these diseases (***Van Goethem et al., 2003***; ***Zeviani et al., 1988***). In the present study, we establish that the region containing '9 bp deletion' can fold into G-quadruplex structures, as shown by EMSA and CD studies in a K+dependent manner. Using bisulphite modification assay, we showed that Region I could fold into a G4 DNA in the mitochondrial genome. Immunofluorescence studies and antibody pull-down assays using G4-specific antibody, BG4, provide further support for



**Figure 9.** Investigation of binding efficacy of Endonuclease G to G4 DNA at Region I of mitochondrial genome. (**A**) Representative image showing colocalization of Endonuclease G with BG4 in HeLa cells. Alexa Fluor 568 and Alexa Fluor 488 conjugated secondary antibodies were used for detection of Endonuclease G and BG4 proteins, respectively. DAPI was used as nuclear stain. (**B**) The quantitation showing colocalization of Endonuclease G and BG4. The colocalization was quantified using Mander's colocalization coefficient (ImageJ software) by analyzing a minimum of 100 cells and presented as a dot plot. Red plot represents the overlapping of Endonuclease G over BG4 while green plot represents the overlapping of BG4 over Endonuclease G. (**C**) Schematic showing the pull-down assay used for evaluation of binding of Endonuclease G present in the rat testicular mitochondrial extracts to the mitochondrial genome. Bound regions were pulled out using anti-Endonuclease G and protein A/G beads. Regions of interest were detected by either semi-quantitative PCR or real-time PCR using appropriate primers. (**D**) Agarose gel profile showing the amplification through semi-quantitative PCR of Input DNA (upper panel) and Endonuclease G pull down DNA (lower panel). Primers specific to 5 G-quadruplex forming regions (GR1-GR5) and 10 random regions (CR1-CR10) were also used for the amplification. (**E**) Real-time PCR of 5 G-quadruplex forming regions (blue) and 10 random regions (green) following pull-down assay. Input DNA served as template control. Antibody control served as a negative control. Error bar represents three independent biological repeats. (**F**) Evaluation of binding of Endonuclease G to different regions of the mitochondrial genome within cells by mito IP. Cells were crosslinked and then mitochondria were isolated. Endonuclease G bound DNA was obtained and was amplified for different regions of mitochondria, as explained in panel E. Graph is plotted for the Ct values obtained following real-time PCR as described above. The error bar represents three independent biological repeats. Refer also *Figure 9—figure supplement 1*, *Figure 2*.

The online version of this article includes the following source data and figure supplement(s) for figure 9:

**Source data 1.** ChIP assay showing the binding of Endonuclease G with the mitochondrial G-quadruplex regions within cells.

**Source data 2.** ChIP assay showing the binding of Endonuclease G with the mitochondrial G-quadruplex regions when purified Endonuclease G was used.

**Source data 3.** P1 nuclease assay showing the binding of Endonuclease G to mitochondrial G quadruples regions.

**Figure supplement 1.** Binding of Endonuclease G to G-quadruplex regions of the mitochondrial genome.

*Figure 9 continued on next page*

**Figure supplement 2.** Binding of Endonuclease G to G-quadruplex regions of the mitochondrial genome.

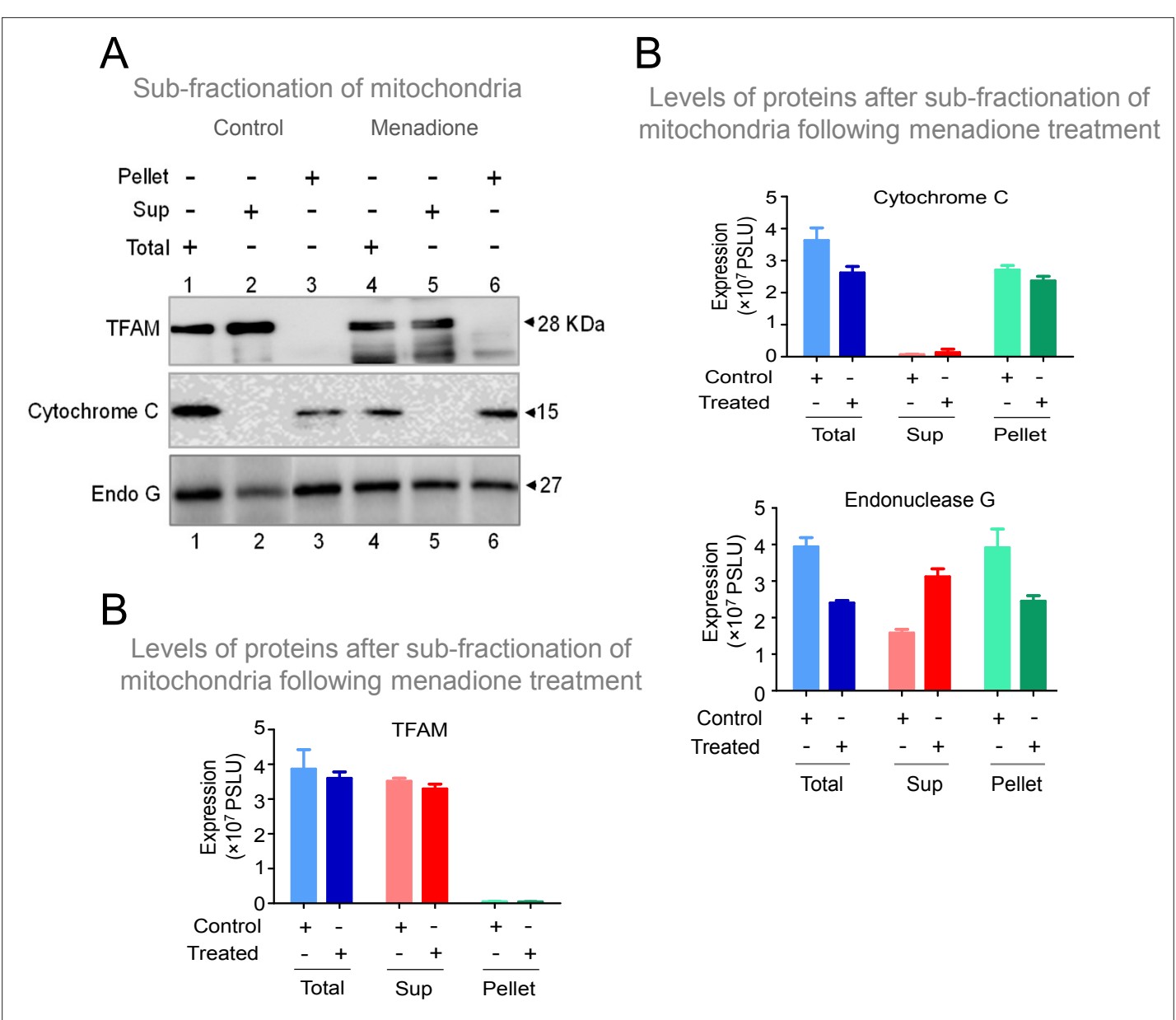

**Figure 10.** Investigation of stress conditions that favor the transport of Endonuclease G to mitochondrial matrix. (**A**) Western blot showing the presence of Endonuclease G, Cytochrome C and TFAM in either total or supernatant (Sup) and pellet fraction with or without menadione treatment (25 µM) following sub-fractionation of mitochondrial compartments. Lanes 1–3 are for control samples and Lanes 4–6 are menadione treated samples. (**B**) Bar graph showing the quantitation of presence of Endonuclease G, Cytochrome C and TFAM in either total mitochondria or supernatant (Sup) and pellet fraction with or without menadione treatment (25 µM). Quantitation is based on three biological repeats and the data is shown with the error bar calculated as SEM. The values in Y-axis are expressed in PSLU (photo-stimulated luminescence units) representing the expression.

The online version of this article includes the following source data for figure 10:

**Source data 1.** Sub localization of Endonuclease G with or without induction of stress.

Figure 11. Model depicting mechanism of generation of '9 bp deletion' seen in the mitochondrial genome. When mitochondria are under stress, Endonuclease G releases into matrix from inner membrane space. In the matrix, Endonuclease G binds and induces cleavage at single-double stranded junctions of G4 DNA as indicated by arrows. Exonuclease action (CtIP/MRE11/Exo G) exposes the direct repeats/microhomology region, which is then paired with the help of unknown proteins, most likely using direct repeats. Ligation of broken ends can result in a '9 bp deletion' at the Region I of the mitochondrial genome.



the occurrence of G4 quadruplex DNA structures within the cells. Thus, with the combinatorial use of in silico studies, and biophysical, biochemical and ex vivo techniques, we demonstrate the formation of G quadruplexes at the region corresponding to a 9 bp deletion in the mitochondrial genome.

The formation of intramolecular parallel G-quadruplex structure in the mitochondrial genome could make the region fragile as there is an increased probability of replication fork slippage or transcription arrest at this region. The primer extension assay revealed multiple pause sites around the G4 motif, suggesting that polymerase arrest could occur at these structures. However, the deletions seen were precise and occurred in a particular sequence. Thus, an alternate possibility for generating a break at these structures could be due to recognition by structure-selective nucleases followed by cleavage at G4 DNA.

Studies have shown that a few enzymes possess the ability to cleave G-quadruplex structures. The action of RAGs at the G-quadruplex site in BCL2 MBR (*Raghavan et al., 2004*), MRE11 and DNA2 helicase on either side of the G-quadruplex structure in vitro (*Masuda-Sasa et al., 2008*) being some of the examples. In the current study, we observed that the structure-selective nuclease activity of mitochondrial endonuclease G could induce cleavage at G4 DNA formed at Region I of the mitochondrial genome using both biochemical (primer extension with purified nucleases, mito-IP with purified Endonuclease G and isolated mitochondria, reconstitution assay with plasmid and purified protein) and ex vivo (immunofluorescence, mito-IP, reconstitution assay in mitochondria) assays. Previous studies have reported various other properties of Endonuclease G, which include the generation of primers during mitochondrial DNA replication by DNA polymerase γ, initiation of genomic inversion in herpes simplex type-1 virus (HSV-1), and its role during apoptosis (*Côté et al., 1989*; *Parrish et al., 2001*; *Ruiz-Carrillo and Renaud, 1987*).

Mammalian Endonuclease G is encoded in the nucleus (*Gannavaram et al., 2008*; *Ohsato et al., 2002*; *Wiehe et al., 2018*; *Zhou et al., 2016*) and upon translocation to mitochondria, the mitochondrial targeting sequence is cleaved off such that a mature nuclease is released (*Li et al., 2001*; *van Loo et al., 2001*; *Wiehe et al., 2018*). Generally, Endonuclease G is known to reside in the intermembrane space of mitochondria, although a low expression has been reported in the matrix (*Duguay and Smiley, 2013*; *McDermott-Roe et al., 2011*; *Wiehe et al., 2018*). Our preliminary results suggest that Endonuclease G may be further released to the matrix when cells are under stress (*Figure 10*).

In a recent study, it has been shown that Endonuclease G also cleaved mtDNA in the case of unrepaired oxidative lesions (*Wiehe et al., 2018*). This was helpful in compensatory mtDNA replication through TWINKLE and mtSSB (*Wiehe et al., 2018*). However, the binding of Endonuclease G to mtDNA and cleavage might be an event dependent upon various factors, including oxidative stress. Therefore, it is possible that in a typical cellular condition, the level of Endonuclease G in the matrix is low; however, upon stress, it is released into the matrix leading to genome fragility. Hence, the stress-induced release may act as a regulatory mechanism concerning the generation of mitochondrial deletions. A recent study showed the complete degradation property of Endonuclease G at pH 6, which could delineate its role during apoptosis, since apoptotic cells show cytosolic acidification (*Schäfer et al., 2004*). Interestingly, at physiological pH, the enzyme showed the property of nuclease (converting supercoiled DNA to open circular and linear; *Schäfer et al., 2004*). CPS-6, a mitochondrial endonuclease G in *Caenorhabditis elegans,* acts with maternal autophagy and proteasome machinery to promote paternal mitochondrial elimination. It relocates from the intermembrane space of paternal mitochondria to the matrix after fertilisation to degrade mitochondrial DNA (*Zhou et al., 2016*). *Endog*[-/-] mice revealed that Endonuclease G could act as a novel determinant of maladaptive cardiac hypertrophy, which is associated with mitochondrial dysfunction and depletion (*McDermott-Roe et al., 2011*).

Our study suggests that Endonuclease G can bind to G-quadruplexes and nick the DNA, resulting in single-strand breaks, which can be converted as double-strand breaks during replication or other DNA transactions. However, in low frequency, we could also see DSBs directly induced by Endonuclease G. We observed that the binding was specific to G4 DNA, as we did not observe nonspecific binding with regions where GC content was high in the mitochondrial genome. Previous studies have shown that although Endonuclease G showed random nuclease activity at higher concentrations, at optimal concentrations, it was specific to kink DNA (*Ohsato et al., 2002*). The first study on Endonuclease G showed preference towards the guanine-rich strands, which falls in line with our study as the formation of G quadruplexes requires a G-rich region (*Ruiz-Carrillo and Renaud, 1987*). Further,

Endonuclease G cleaved mtDNA in unrepaired oxidative lesions (*Wiehe et al., 2018*). It is likely that binding and cleavage by Endonuclease G on mtDNA be an event dependent on multiple factors, including oxidative stress. Based on our data, it is evident that in the presence of stress, levels of Endonuclease G go up in the matrix, which enables the protein to be present proximal to the mitochondrial genome. The findings from mito IP studies suggest that Endonuclease G only binds to the G-quadruplex forming regions and not to the other areas of mtDNA. This provides additional regulation and ensures the mitochondrial genome stability in most instances.

DNA repair in mitochondria is poorly understood. Base excision repair (BER) is the most well-characterized DNA repair pathway in mitochondria, while it is generally believed that nucleotide excision repair is absent (*Akbari et al., 2008*; *de Souza-Pinto et al., 2009*; *Liu et al., 2008*). Homologous recombination repair helps repair DSBs and, thus stability of the genome (*Dahal et al., 2018*), though classical NHEJ was undetectable (*Tadi et al., 2016*). In contrast, microhomology-mediated end joining uses flanking direct repeats to fix DSBs and delete the sequences between the repeats (*Tadi et al., 2016*). Interestingly, the G quadruplex forming Region I is close to the deletion junction 8271:8281 (*Redd et al., 1995*; *Yao et al., 2000*) also had a 9 bp direct repeat sequence at the flank site.

In a recent study, a replication slippage-mediated mechanism was attributed to a common 4977 deletion where in frequent replication fork stalling at the junction of the common deletion was observed (*Phillips et al., 2017*). While this is a replication-dependent process, what we observed may not be immediately connected to replication. However, it is possible that replication can indeed facilitate the formation of G-quadruplexes.

A balance between the formation and resolution of G4 DNA within the mitochondrial genome is essential. The mitochondrial genes are transcribed as a single polycistronic mRNA (*Clayton, 1984*; *Sanchez et al., 2011*) and therefore undergo a high transcription rate, which may promote G-quadruplex formation. Besides, random 'breathing' of DNA (a process of transient melting of the duplex structure due to thermal fluctuations) may also occur in the GC-rich sequences of the mitochondrial DNA promoting G-quadruplex formation (*Dornberger et al., 1999*; *Jose et al., 2009*). Once the structure is formed, G4 resolvases also play a critical role in its resolution. In the nuclear DNA context, resolvases like WRN and BLM helicase are considered as the factors that negatively regulate G4 DNA (*Giri et al., 2011*; *Paeschke et al., 2013*). However, such a role for these nucleases in mitochondria is not reported. TWINKLE helicase is known to unwind mitochondrial duplex DNA and regulate replication, although its role in structural resolution is yet to be studied (*Phillips et al., 2017*).

Hence, our study suggests a novel mechanism that explains fragility in mitochondria, dependent on the structure-selective nuclease activity of Endonuclease G on G-quadruplex DNA. The process could be controlled by cellular stress conditions, which may regulate the release of Endonuclease G to the mitochondrial matrix; however, this requires further extensive investigation. Although we have focused on a single example of '9 bp deletion' often seen in mitochondria, it is very much possible that it may be a general mechanism and may help in explaining several other deletions seen in mitochondria-associated human disorders.

## Materials and methods
### Enzymes, chemicals, and reagents
Chemicals and reagents used in the present study were purchased from Millipore-Sigma (USA), SRL (India), Himedia (India), and Amresco (USA). Restriction enzymes and other DNA-modifying enzymes were obtained from New England Biolabs (USA). Culture media was from Sera Laboratory International Limited (UK), Lonza (UK). Fetal bovine serum and PenStrep were from Gibco BRL (USA). Radioisotope-labelled nucleotides were from BRIT (India). Antibodies were purchased from Abcam (UK), Santa Cruz Biotechnology (USA), BD (USA), Cell Signaling Technology (USA) and Calbiochem (USA). Oligomeric DNA was purchased from Juniper Life Sciences (India) and Medauxin (India).

### Oligomeric DNA
The oligomers used in this study are listed in *Table 1*. Oligomeric DNA was purified on 12–15% denaturing PAGE when needed (*Nambiar and Raghavan, 2012*; *Raghavan et al., 2005b*; *Tadi et al., 2016*). The details of oligomers used for different experiments are provided in appropriate sections.

**Table 1.** Oligomers used in the study.
Refer also *Figure 1*, *Figure 2*, *Figure 3*, *Figure 5*.

| Oligomer Name | Sequence | Region |
|---|---|---|
| VKK11 | 5'- GCTGTGTCGACTACTACGGTCAATGCTCTG –3' | GR1 |
| VKK12 | 5'- CTGAGGTCGACTGGGTGATGAGGAATAGTG - 3' | |
| RBK46 | 5'- TAATCAACACCCTCCTAGCC –3' | GR2 |
| VKK14 | 5'- GATAGTGTCGACGGCTCATGGTAGGGGTAA –3' | |
| SD10 | 5'- TTCGCTGACGCCATAAAACT –3' | GR3 |
| SD11 | 5'- ATCAGGGCGTAGTTTGA –3' | |
| SD12 | 5'- GCTCACAAGAACTGCTAA –3' | GR4 |
| SD13 | 5'- TGGATGCGACAATGGAT –3' | |
| SD14 | 5'- TCTTGCACGAAACGGGAT –3' | GR5 |
| SD15 | 5'- TAGGATGAGGATGGATAGT –3' | |
| RBK15 | 5'- CTACTCCTGCTCGCATCTGC –3' | CR2 |
| RBK16 | 5'- GAAGGTGGTGTTGAGGTTGC –3' | |
| RBK17 | 5'- GCATTGTTCGTTACATGGTCC –3' | CR3 |
| RBK18 | 5'- GTGGAAGCGGATGAGTAAGAAG –3' | |
| RBK19 | 5'- CTCACCACTACAATCTTCCTAG –3' | CR4 |
| RBK20 | 5'- CAAAGATGGTAGAGTAGATGACG –3' | |
| RBK21 | 5'- CTAACCATCTTCTCCTTACACCTAG –3' | CR5 |
| RBK22 | 5'- GTTTGCTAATACAATGCCAGTCAGG –3' | |
| RBK23 | 5'- CGAAGGTGGATTTAGCAGTAAACTG –3' | CR6 |
| RBK24 | 5'- CGGTACTATATCTATTGCGCCAGG –3' | |
| RBK41 | 5'- GTATCATCAACTGATGAGCAAG –3' | CR1 |
| SD2 | 5'- TCAGCAAACCCTGATGAA –3' | CR7 |
| SD3 | 5'- CACTCTACTCTCAGTTTACT –3' | |
| SD4 | 5'- ACATCGAATACGCCGCA –3' | CR8 |
| SD5 | 5'- AGTTGGTCGTAGCGGAATCG –3' | |
| SD6 | 5'- TAGGGTTTATCGTGTGAG –3' | CR9 |
| SD7 | 5'- AGTGTGGCGAGTCAGCT –3' | |
| SD8 | 5'- TACTCACTCTCACTGCCCAA –3' | CR10 |
| SD9 | 5'- TGTTTGTCGTAGGCAGAT-3' | |
| VKK21 | 5'- GGATCCATGCGGGCGCTGCGG –3' | |
| VKK22 | 5'- GCGGCCGCTCACTTACTGCCCG –3' | |
| DI12 | 5'- GCAAACCACAGTTTCATGCCCATC –3' | |
| DI13 | 5'- GCCTATAATCACTGCGCCCGCTC –3' | |
| VKK1 | 5'- CCCGTATTTACCCTATAGCACCCCCTCTACCCCC –3' | C1 |
| VKK2 | 5'- GGGGGTAGAGGGGGTGCTATAGGGTAAATACGGG –3' | G1 |
| VKK5 | 5'- GTCAGTAGAGGGGGTGCTATAGGGTAAATACGGG –3' | M1 |
| VKK6 | 5'- GTCAGTAGAGAATGTGCTATAGGGTAAATACGGG –3' | M2 |
| VKK7 | 5'- GTCAGTAGAGGGGGTGCTATATCATAAATACGGG –3' | M3 |
| SD 54 | 5'- GGCCAGGGCCCCGCGGTCGAAGCCACTGCC-3' | |
| SD 57 | 5'- TCACCTGGCCGCCGCCGCCAACCAC-3' | |

*Table 1 continued on next page*

*Table 1 continued*

| Oligomer Name | Sequence | Region |
|---|---|---|
| DK27 | 5'-TGGGCTCTAGAGGACATAGAGTAAGTGCT-3' | |
| DK28 | 5'-AGCACTTACTCTATGTCCTCTAGAGCCCA-3' | |
| KD14 | 5'-CAAGCTCGAAATTAACCCTCAC-3' | |
| KD13 | 5'-CCCAGTCACGACGTTGTAAAAC-3' | |
| DI8 | 5'-CTTACAGTGGGCTCTAGAGGGGGTAGATAATAT GCTATAGGGTAAATACTCACTAAAAATCTTTGAA-ATAGGG –3' | |
| DI9 | 5'-CTAAAAATCTTTGAAATAGGGTGAGTATTTA CCCTATAGCATATTATCTACCCCCTCTAGAGCCCA-CTGTAAG –3' | |

A more simplified nomenclature is used for certain oligomers while explaining the results of respective experiments.

## Cell lines

HeLa (human cervical cancer), HCT116 (human colon cancer), MEF (mouse embryonic fibroblast) and HEK 293T (human embryonic kidney epithelial cell line) were purchased from National Centre for Cell Science, Pune, India. Nalm6 and Reh cells were from Dr M. R. Lieber (USA), and Rho(0) cells were from J. Neuzil (Australia). The identity of all these cell lines was confirmed by performing STR profiling (DNA labs India, Hyderabad, India). All the cell lines were found to be free of mycoplasma contamination. The Mycoplasma test was conducted in the laboratory using the MYCOseq kit (Thermo Fisher Scientific, Waltham, Massachusetts) and further validated by DNA labs India, Hyderabad, India. Cells were cultured in RPMI1640 or MEM medium supplemented with 10–15% fetal bovine serum (FBS), 100 µg/ml Penicillin, and 100 µg/ml streptomycin and incubated at 37 °C in a humidified atmosphere containing 5% $CO_2$ as described before (*Ghosh et al., 2022*; *Kumari et al., 2021*). Rho(0) cells of B16 cells origin were cultured in DMEM medium supplemented with 15% fetal bovine serum (FBS), 100 µg/ml Penicillin, 100 µg/ml streptomycin, 50 mg/ml Uridine and 1 mM Sodium pyruvate and incubated as described above (*Dong et al., 2017*; *Tan et al., 2015*).

## Animals

Male Wistar rats (*Rattus norvegicus*) 4–6 weeks old were purchased from the Central animal facility, Indian Institute of Science (IISc), Bangalore, India, and maintained as per the guidelines of the animal ethical committee in accordance with Indian National Law on animal care and use (CAF/Ethics/526/2016).

## shRNA

shRNA used in the study (TRCN0000039643) was purchased from the Resource Center at Division of Biological Science, IISc, Bangalore (funded by DBT: BT/PR4982/AGR/36/718/2012) Indian Institute of Science, Bangalore (India).

## Plasmid constructs

### Plasmid construction for mitochondrial G-quadruplex assays

Mitochondrial DNA isolated from Nalm6 cells was amplified using primers specific for Region I (VKK11 and VKK12). The PCR products were purified, blunted and ligated into the EcoRV site of pBlueScript SK(+) to obtain the wild-type plasmid for mitochondrial Region I (pDI1). Mutant plasmids for mitochondrial Region I were generated by PCR-mediated site-directed mutagenesis using pDI1 as a template described before (*Raghavan et al., 2005c*) using the primers VKK11, DI8, VKK12 and DI9 followed by cloning into pBlueScript SK(+). The resulting plasmid, pDI2, had a mutation in two G stretches of Region I. A second mutant plasmid, pDR4 with mutations in three G stretches, was generated by

site-directed mutagenesis using pDI2 as a template using the primers DK27, KD13, KD14, and DK28 followed by cloning into pBlueScript SK(+).

## Construction of Endonuclease G containing vector for purification of the protein

The endonuclease G coding sequence was amplified using primers VKK21 and VKK22, which were designed with BamHI and NotI restriction sites, respectively, at flanking regions. PCR product was gel purified and cloned into the EcoRV site of pBS SK +cloning vector. For the generation of pRBK2, the coding DNA sequence was digested from the cloning vector using BamHI and NotI enzymes and cloned into the same site (BamHI and NotI) of pET28a+. Mutant plasmid for Endonuclease G was generated by PCR mediated site-directed mutagenesis as described before (*Raghavan et al., 2005c*) using the primers VKK21, SD54, VKK22 and SD57 followed by cloning into BamHI and NotI site of pRBK2. The resulting plasmid, pDR1, had a mutation in the nuclease domain of Endonuclease G (Histidine to Alanine at position 141).

## Plasmid isolation and purification

For each plasmid, after transformation, *Escherichia coli* was cultured in 500 ml of Luria broth (HiMedia, USA) for 18 h at 37 °C. Isolation of plasmid DNA was performed by the standard alkaline lysis method. It was then purified and precipitated, as described previously (*Sambrook et al., 1989*). The pellet was dissolved in TE (pH 8.0).

## Isolation of mitochondrial DNA

Mitochondria were isolated from Nalm6 cells as described before (*Dahal et al., 2018*; *Tadi et al., 2016*). Cells were first homogenized in the buffer [70 mM sucrose, 200 mM mannitol, 1 mM EDTA, and 10 mM HEPES (pH 7.4), 0.5% BSA per g tissue] in a Dounce-type homogenizer (20–30 strokes). Homogenate was centrifuged (3000 rpm, 10 min at 4 °C) to remove nuclei and cellular debris from the supernatant that contains the cytosol and mitochondria. The supernatant fraction was centrifuged at 12,000 rpm (30 min at 4 °C) to pellet the mitochondria. The mitochondrial pellet was washed in suspension buffer [10 mM of Tris–HCl (pH 6.7), 0.15 mM of $MgCl_2$, 0.25 mM of sucrose, 1 mM PMSF and 1 mM DTT] twice, and the mitochondrial pellet was incubated in the buffer containing 75 mM NaCl, 50 mM EDTA (pH 8.0), 1% SDS and 0.5 mg/ml Proteinase K to remove denatured proteins. The resulting lysate was deproteinised by phenol-chloroform extraction. The mitochondrial DNA was precipitated with 3 M Sodium acetate (pH 5.2) and chilled with 100% ethanol. The purity of the mitochondrial DNA preparation was evaluated by PCR amplification with nuclear and mitochondrial DNA-specific primers.

## Preparation of mitochondrial extracts

Mitochondrial protein extracts were prepared following the isolation of mitochondria based on differential centrifugation as described (*Dahal et al., 2018*; *Tadi et al., 2016*). Testes and spleen from 4- to 6-week-old male Wistar rats were minced on ice and lysed in mitochondrial lysis buffer [50 mM of Tris–HCl (pH 7.5), 100 mM of NaCl, 10 mM of $MgCl_2$, 0.2% Triton X-100, 2 mM of EGTA, 2 mM of EDTA, 1 mM of DTT and 10% glycerol] by mixing (30 min at 4 °C) along with protease inhibitors PMSF (1 mM), aprotinin (1 µg/ml), pepstatin (1 µg/ml) and leupeptin (1 µg/ml). The mitochondrial extract was centrifuged at 12,000 rpm (5 min), and the supernatant (mitochondrial fraction) was aliquoted, snap-frozen and stored at −80 °C till use.

Mitochondrial protein extracts were also prepared from rat tissues as per the manufacturer's instructions using a mitochondrial extraction kit (Imgenex, USA). The purity of the mitochondrial extracts prepared was checked by immunoblotting using the nuclear and mitochondrial specific markers.

## Fractionation of mitochondria

A total of $4 \times 10^7$ HeLa cells were treated with 25 µM Menadione and incubated for 2 hr at 37 °C. The mitochondrial pellet was first sonicated after incubation in hypotonic buffer 10 mM HEPES-KOH pH7.6, 100 mM NaCl, 10 mM Magnesium acetate and 1 mM phenylmethyl sulfonyl fluoride (PMSF) for 15 min. The same buffer containing 500 mM NaCl was added and incubated for another 15 min and then sonicated for 10 min at 37% duty per cycle. The samples were then separated into membrane

and soluble fractions by ultracentrifugation at 100,000 g (TLA 110 rotor) for 30 min at 4 °C in Opti-maTM TLX table-top ultracentrifuge (Beckman-Coulter) (*Sinha et al., 2010*; *Suzuki et al., 2002*). The supernatant fraction was transferred to a new tube. To the pellet, 1 X PBS was added, and both fractions were used for western blotting analysis as described before (*Chiruvella et al., 2012*).

## 5' end labelling of oligomeric substrates

The 5' end labelling of oligomeric substrates was done using γ[$^{32}$P]-ATP in the presence of 1 U of T4 Polynucleotide kinase in a buffer containing 20 mM Tris-acetate [pH 7.9], 10 mM magnesium acetate, 50 mM potassium acetate and 1 mM DTT at 37 °C for 1 h. The radiolabelled oligomers were purified on a Sephadex G25 column and stored at –20 °C until further use (*Kumar et al., 2010*; *Sharma et al., 2011*; *Tadi et al., 2016*).

## Gel mobility shift assays

Gel mobility shift assays were performed using 5' radiolabelled oligomeric DNA substrates (4 nM) predicted to form G-quadruplex structures, its complementary oligos and the mutants by incubating in a buffer containing 10 mM Tris-HCl (pH 8.0), 1 mM EDTA, with or without 100 mM KCl at 37 °C and resolved on 15% native polyacrylamide gels (*Dahal et al., 2022*; *Nambiar et al., 2011*; *Nambiar et al., 2013*). In conditions where the effect of KCl was studied, 100 mM KCl was added while preparing gels and the running buffer (1 X TBE). Electrophoresis was carried out at 150 V at RT. The dried gels were exposed, and signals were detected by phosphorImager FLA9000 (Fuji, Japan).

For BG4 binding assay, 5' radiolabelled oligomeric DNA substrates (4 nM) predicted to form G-quadruplex structures and its complementary oligomer was incubated in a buffer containing 10 mM Tris-HCl (pH 8.0), 1 mM EDTA, with 100 mM KCl at 37 °C for 1 h (*Das et al., 2016*; *Javadekar et al., 2020*). It was then incubated along with increasing concentrations of purified BG4 protein (100, 200, 400, and 800 ng) in a buffer containing 250 mM Tris (pH 8.0), 1 mM EDTA, 0.5% Triton X-100, 500 μg/ml BSA and 20 mM DTT for 1 hr at 4 °C. The complex was then resolved on 5% native polyacrylamide gels in the presence of 100 mM KCl in gel and the running buffer (1 X TBE). Electrophoresis was carried out at 100 V at 4 °C. The dried gel was exposed to a PI cassette, and the radioactive signals were detected by phosphorImager FLA9000 (Fuji, Japan).

## Circular dichroism (CD)

Circular dichroism studies for analysis of mitochondrial G-quadruplex formation were performed with oligonucleotide substrates (2 μM) corresponding to G-rich sequences and complementary C-rich sequences in a buffer containing 10 mM Tris-HCl (pH 8.0) and 1 mM EDTA. The experiments were set up either in the presence or absence of 100 mM KCl. The spectra were recorded between wavelengths 200 and 300 nm (5 cycles, scan speed of 50 nm/s, RT) using JASCO J-810 spectropolarimeter (*Nambiar and Raghavan, 2012*; *Raghavan et al., 2005a*) and analysed using SpectraManager (JASCO J-810 spectropolarimeter).

## DMS protection assay

The radiolabelled oligomer VKK19 was incubated with Dimethyl Sulphate (1:250 dilution) in buffer containing 10 mM Tris-HCl (pH 8.0) and 1 mM EDTA either in the presence or absence of 100 mM KCl, at RT for 15 min (*Kumari et al., 2019*; *Nambiar et al., 2013*). An equal volume of 10% piperidine was added, and the reactions were incubated at 90 °C for 30 min. The reaction mixtures were diluted 2-fold with double distilled water and vacuum dried in a Speed Vac concentrator. The resulting pellet was again washed with water and vacuum dried. This procedure was repeated thrice. The pellet was finally resuspended in 10 μl TE buffer (10 mM Tris-HCl, (pH 8.0) and 1 mM EDTA), mixed with formamide containing dye, and resolved on a 15% denaturing PAGE.

## Polymerase stop assays

Plasmid substrates pDI1, pDI2 (mitochondrial Region I), Vent Exo(-) polymerase was used for primer extension assays (*Kumari et al., 2015*; *Nambiar et al., 2013*) in a buffer containing 20 mM Tris-HCl (pH 8.8), 10 mM (NH$_4$)$_2$SO$_4$, 10 mM KCl, 2 mM MgSO$_4$, 0.1% Triton X-100 with additional supplementation with 75 mM KCl, LiCl or NaCl where required. After the addition of 200 μM dNTPs, 0.2 U of Vent Exo(-) and 5' radiolabelled primers (VKK11, VKK12, DI12), the reaction was carried out in a one-step

PCR-mediated primer extension assay for mitochondrial Region I (95 °C for 10 min, 55–65°C for 3 min as specified below for each primer and 75 °C for 20 min, as a single cycle) or multi-cycle PCR extension (Denaturation at 95 °C for 5 min, followed by 95 °C for 45 s, 55 °C for 45 s and 72 °C for 45 s for 15 cycles and a final extension at 72 °C for 5 min). The reaction products were loaded on 8% denaturing PAGE, and the products were visualised as described above.

## Endonuclease cleavage assay

Wild-type plasmid (pDI1) or mutant plasmid (pDI2) or mitochondrial DNA was incubated with mitochondrial extract or purified Endonucleases (CtIP, FEN1, RAG, Endonuclease G [wildtype or mutant]) in a buffer containing 25 mM MOPS (pH 7.0), 30 mM KCl, 30 mM potassium glutamate and 5 mM $MgCl_2$ at 37 °C for 1 hr. In the control reaction, only DNA and buffer were added. Following termination of the reaction, purified samples were loaded on 0.8% agarose gel or subjected to primer extension assay using a radiolabeled primer (VKK11 or VKK12) and resolved on 8% denaturing PAGE.

## Preparation of sequencing ladders

Sequencing ladders for the primers VKK11 and VKK12 were prepared using the cycle sequencing method with a dNTP:ddNTP ratio of 1:20, 1:40, 1:30 and 1:10 for C, T, A and G ladders, respectively. The dNTP:ddNTP mix were separately provided in reaction mixtures containing 10 nM plasmid template, 0.5 µM 5' radiolabelled primers, 20 mM Tris-HCl (pH 8.8), 10 mM $(NH_4)_2SO_4$, 10 mM KCl, 2 mM $MgSO_4$, 0.1% Triton X-100 and 1 U of Vent Exo (-) polymerase. The PCR was carried out using the following conditions: Step 1 at 95 °C for 30 s, 60 °C for 30 s and 72 °C for 1 min (25 cycles), followed by Step 2 at 95 °C for 30 s and 72 °C for 2 min (10 cycles).

## Sodium bisulphite modification assay

pDI1, plasmid containing mitochondrial Region I or mitochondrial DNA isolated from Nalm6 cells, were treated with sodium bisulfite as described earlier (*Raghavan et al., 2004*; *Raghavan et al., 2006*). Briefly, approximately 2 µg of mitochondrial DNA was treated with 12.5 µl of 20 mM hydroquinone and 458 µl of 2.5 M sodium bisulfite (pH 5.2) at 37 °C for 16 hr. The bisulphite-treated DNA was purified using Wizard DNA Clean-Up Kit (Promega, Madison, WI) and desulfonated by treating with 0.3 M NaOH (15 min at 37 °C). The DNA was ethanol-precipitated and resuspended in a 20 µl TE buffer. The mitochondrial Region I was PCR amplified, resolved on 1% agarose gels, purified, and TA cloned. Clones were sequenced to analyse for conversions. The experiment was repeated three independent times, and the cumulative data is presented.

## Overexpression and purification of endonucleases

### Endonuclease G and its mutant

For protein expression, *E. coli* Rosetta cells were transformed with pRBK2 or pDR1, grown until OD reaches 1.0 and then induced with 1 mM IPTG at 16 °C for 16 hr. Cells were harvested, and the extract was prepared using extraction buffer (20 mM Tris-HCl (pH 8.0), 0.5 M KCl, 20 mM imidazole (pH 7.0), 20 mM Mercaptoethanol, 10% glycerol, 0.2% Tween 20, 1 mM PMSF) by sonication followed by loading on to Ni-IDA column (Macherey-Nagel, Germany). Endonuclease G or the mutant was eluted in gradient imidazole concentration (100–500 mM) and pure fractions were pooled, and the identity of the protein was confirmed by immunoblotting and used for the assays.

### RAGs

MBP cRAGs (RAG1, amino acids 384–1040; RAG2, amino acids 1–383) were purified using a method as described previously (*Naik et al., 2010*; *Raghavan et al., 2005b*). Briefly, 293T cells were transfected with 10 µg of plasmid by the calcium phosphate method. After 48 hr of transfection, cells were harvested, and proteins were purified using an amylose resin column (New England Biolabs). Fractions were eluted and checked by silver staining. The activity was and studied by nicking assay on standard recombination signal sequence substrate (AKN1/2; *Nishana and Raghavan, 2012*).

### FEN1

FEN1 was purified using the expression plasmid pET-FEN1 CH described previously (*Greene et al., 1999*). Briefly, culture was grown until OD reached ~0.5, induced with 0.5 mM IPTG at 37 °C for 3 hr.

Cells were harvested, lysed, and the lysate was loaded onto the Ni-NTA column. FEN1 was eluted in increasing imidazole concentrations (100–500 mM), and pure fractions were pooled, and the identity of the protein was confirmed by immunoblotting and used for the assays.

## CtIP

CtIP was purified using the expression plasmid pET14b-CtIP as described previously (*Yu and Baer, 2000*). Briefly, culture was grown till OD reached 0.6, induced with 0.5 mM IPTG at 37 °C for 4 hr. Cells were harvested, lysed, and the lysate was loaded on to Ni-NTA column. FEN1 was eluted in increasing imidazole concentrations (100–800 mM), pure fractions were pooled, and the protein's identity, was confirmed by immunoblotting and used for the assays.

## Overexpression and purification of BG4

The plasmid expressing BG4 protein, pSANG10-3F-BG4, was a gift from Shankar Balasubramanian (Addgene plasmid # 55756). The plasmid was transformed into *E. coli*, BL21 (DE3), and the culture was expanded by incubating at 30 °C till the O.D. reached up to 0.6 (*Biffi et al., 2013*; *Das et al., 2016*; *Javadekar et al., 2020*). The cells were then induced with 1 mM IPTG for 16 hr at 16 °C, harvested, and resuspended in lysis buffer (20 mM Tris-HCl [pH 8.0], 50 mM NaCl, 5% glycerol, 1% Triton X-100 and 1 mM PMSF). The cells were lysed by sonication, centrifuged, and the supernatant was then loaded onto a Ni-NTA chromatography column (Novagen, USA). BG4 was eluted using increasing concentrations of Imidazole (100–400 mM). BG4 enriched fractions were dialysed against dialysis buffer (PBS containing 0.05% Triton X-100, 1 mM Mercaptoethanol, 5% glycerol and 0.1 mM PMSF) overnight at 4 °C. The identity of the protein was confirmed by immunoblotting using an anti-FLAG antibody (Calbiochem, USA) (*Kumari et al., 2019*). Activity assay for each batch of BG4 was checked by performing a binding assay as described previously (*Javadekar et al., 2020*).

## Immunoblot analysis

For immunoblotting analysis, approximately 20–30 µg protein was resolved on 8–10% SDS-PAGE (*Chiruvella et al., 2008*; *Kumar et al., 2010*; *Thomas et al., 2016*). Following electrophoresis, proteins were transferred to the PVDF membrane (Millipore, USA), blocked using 5% non-fat milk or BSA in PBS with 0.1% Tween-20. Proteins were detected with appropriate primary antibodies against Cytochrome-C (ab90529), PCNA (SC56), Tubulin (SC5286), CtIP (SC22838), FEN1 (SC13051), RAG1 (SC5599), Endonuclease G (#4969), TFAM (ab176558) and appropriate secondary antibodies as per standard protocol. The blots were developed using a chemiluminescent substrate (Immobilon western, Millipore, USA) and scanned by a gel documentation system (LAS 3000, FUJI, Japan).

## Immunoprecipitation (IP)

IP experiments were performed with modifications (*Chiruvella et al., 2012*; *Sharma et al., 2015*; *Totaro et al., 2011*; *Yeretssian et al., 2011*). Protein A agarose beads (Sigma) were activated by immersing in water and incubated in IP buffer (300 mM NaCl, 20 mM Tris-HCl (pH 8.0), 0.1% NP40, 2 mM EDTA, 2 mM EGTA, and 10% glycerol) for 30 min on ice following which the beads were conjugated with the appropriate antibody at 4 °C for overnight to generate antibody-bead conjugate. The antibody-bead conjugates were then separated by centrifugation and incubated with rat tissue mitochondrial extracts at 4 °C overnight. Then the conjugate bound to the target proteins was separated and washed. Immunoblot analysis confirmed the protein depletion in the resulting supernatant and quantified using Multi Gauge (V3.0). This immunodepleted extract was used for cleavage assay.

## Knockdown of Endonuclease G within cells

HeLa cells (10X10[5]) were seeded in a culture petridish. 10 µg of Endonuclease G shRNA plasmid was transfected using linear/branched PEI (Polyethylenimine) polymer (Sigma, 1 mg/ml) (*Kumari et al., 2021*; *Longo et al., 2013*; *Raymond et al., 2011*; *Srivastava et al., 2012*). Post transfection (48 hr), cells were harvested, and mitochondrial extracts were prepared by differential centrifugation as described above. The knockdown was confirmed using western blotting. Cleavage assay on pDI1 following primer extension was performed using the knockdown extracts. The scrambled plasmid was also used for transfection, which acted as a control for the experiment.

## Binding of Endonuclease G to mitochondrial DNA

5 µg of mitochondrial DNA was incubated with either 5 µg purified Endonuclease G or 50 µg of mitochondrial extract in a 100 µl reaction in PBS. DNA-protein interaction was crosslinked by using 0.1% formaldehyde at 37 °C for 10 min, following quenching the reaction with glycine (125 mM) for 5 min. The complex was sonicated (VC 750, Ultrasonic processor) for 10 min with the pulse of 10 sec on/ 10 sec off and incubated at –80 °C for 8 h. 75 ng of Endonuclease G antibody (CST: 4969) was added and incubated at 4 °C (end to end rotation). A total of 15 µl protein A/G beads (Santa cruz, sc-2003) were added to the chromatin-antibody complex and incubated for 2 h. The supernatant was removed entirely, and beads were washed and eluted with high salt immune complex wash buffer (50 mM HEPES (pH 7.9), 500 mM NaCl, 1 mM EDTA, 0.1% SDS, 1% Triton X-100, 0.1% deoxycholate) and incubated at 55 °C for 2 h. DNA protein interactions were reverse crosslinked by incubating the samples overnight at 65 °C. DNA was purified with phenol: chloroform extraction followed by ethanol precipitation (*Carey et al., 2009*). Endonuclease G pulled-down mitochondrial DNA and was subjected to semiquantitative and real-time PCR of different mitochondrial regions (five G-quadruplex forming regions and ten random control regions). Three independent reactions were performed, and the bar diagram represented threshold values.

## Mitochondrial IP (mito IP)

$4 \times 10^7$ Nalm6 cells were crosslinked with 1% formaldehyde at 37 °C for 15 min and quenched with 100 µl/ml of 1.375 M glycine. Mitochondria were then isolated from the cells as described above and lysed using buffer (5 mM PIPES, 85 mM KCl, 0.5% NP40). Lysed samples were then sonicated (Diagenode Bioruptor, Belgium) with 30 s on/ 45 s off pulse for 30 cycles. Sonicated samples were stored at –80 °C for at least 8 hr (*Carey et al., 2009*). For immunoprecipitation, the samples were centrifuged at high speed for 15 min, and the supernatant was divided for input, secondary control and experimental. Relevant antibodies (Endonuclease G or BG4) were added to the sample and allowed to bind for 8–10 hr. Protein A/G-agarose beads (Sigma, USA) were added to the models and incubated for 2 hr. Samples were washed and eluted using high salt wash buffer (50 mM HEPES (pH 7.9), 500 mM NaCl, 1 mM EDTA, 0.1% SDS, 1% Triton X-100, 0.1% deoxycholate) in repeated cycles of centrifugation and then reverse cross-linked by incubating overnight at 65 °C. Finally, DNA was purified following phenol: chloroform extraction and precipitation. The purified DNA was used for real time PCR amplification of different mitochondrial regions that either support formation of G-quadruplex structure or do not support any secondary structure formation.

## Immunofluorescence

Immunofluorescence studies were performed as described before (*Dahal et al., 2018*; *Kumari et al., 2019*; *Ray et al., 2020*; *Tadi et al., 2016*). Approximately 50,000 cells (HeLa, HEK293T, Rho(0), or MEF) were grown in the media for 24 hr, as described before. Cells were grown in chamber slides in MEM medium supplemented with 10% FBS and 1% penicillin-streptomycin (Sigma) for 24 hr. The cells were stained with either 100 ng/ml of MitoTracker Red 580 (Invitrogen) or 100 ng/ml MitoTracker Green FM (Invitrogen) at 37 °C in a $CO_2$ incubator for 30 min (*Tadi et al., 2016*). Cells were washed twice with 1 X PBS, fixed in 2% paraformaldehyde (20 min) and permeabilised with 0.1% Triton X-100 (5 min) at room temperature. BSA (0.1%) was used for blocking (30 min), and subsequently the cells were incubated with the appropriate primary antibody at room temperature (4 hr). Appropriate FITC or Alexa Fluor conjugated secondary antibodies were used to detect the signal. After washing, the cells were stained with DAPI, mounted with DABCO (Sigma) and imaged under a confocal laser scanning microscope (Zeiss LSM 880 with ×63 magnifications or Olympus FLUOVIEW FV3000 with ×100 magnifications). The images were processed using either Zen Lite or FV31S-SW software.

## Statistical analysis

Every experiment was repeated multiple times with at least three biological repeats and presented. Based on the data obtained from repeats, Student's t-test (two-tailed) was performed, and the error bars were calculated for the bar diagrams representing were the fold change value using GraphPad Prism ver 5.0. Error bars are shown depicting mean ± SEM (ns: not significant, *p<0.05, **p<0.005, ***p<0.0001). Colocalisation analysis in immunofluorescence assay was done using JaCoP in ImageJ

software. The values were plotted in GraphPad Prism 5.0, and the significance was calculated using the same. The values were obtained for at least 50–100 cells per sample.

## Acknowledgements

We thank Bibha Choudhary, Mridula Nambiar, Urbi Roy, Nitu Kumari, and other members of the SCR laboratory for critical reading and comments on the manuscript. We thank Monica Pandey, Rupa Kumari, Mahesh Hegde, Meghana Manjunath and Supriya Vartak for their technical help. We also thank J Neuzil (Australia), A Agarwal (India) and N Bhatraju for support with Rho(0) cells, S Balasubramanian (UK) for pSANG10-3F-BG4 plasmid, Michael A Resnick (USA) for FEN1 plasmid, Richard Baer (USA) for CtIP-pET14b plasmid, Patrick Swanson (USA) for MBP-RAG constructs, Kumar Somasundaram (India) for pIRES2-EGFP and Arun Kumar (India) for pcDNA3.1RFP. We thank Confocal, FACS and Central Animal Facilities, shRNA core facility of IISc, for their help. This work was supported by grants from CSIR (37 (1692)/17/EMR-11), DAE (21/01/2016-BRNS/35074), DBT-COE (BT/PR/3458/COE/34/33/2015), IISc-DBT partnership program [BT/PR27952-INF/22/212/2018] to SCR. SD is supported by a Senior Research Fellowship (SRF) from IISc, and HS is supported by a Junior Research Fellowship (JRF) from CSIR.

## Additional information

### Funding

| Funder | Grant reference number | Author |
|---|---|---|
| Council of Scientific and Industrial Research, India | 37(1692)/17/EMR-11 | Sathees C Raghavan |
| Department of Atomic Energy, Government of India | 21/01/2016-BRNS/35074 | Sathees C Raghavan |
| Department of Biotechnology, Ministry of Science and Technology, India | BT/PR/3458/COE/34/33/2015 | Sathees C Raghavan |
| IISc-DBT partnership programme | BT/PR27952-INF/22/212/2018 | Sathees C Raghavan |
| Indian Institute of Science | | Sumedha Dahal Humaira Siddiqua |

The funders had no role in study design, data collection and interpretation, or the decision to submit the work for publication.

### Author contributions

Sumedha Dahal, Conceptualization, Formal analysis, Validation, Investigation, Methodology, Writing - original draft, Writing – review and editing; Humaira Siddiqua, Formal analysis, Validation, Investigation, Methodology, Writing - original draft; Shivangi Sharma, Ravi K Babu, Sheetal Sharma, Investigation, Methodology; Diksha Rathore, Sathees C Raghavan, Conceptualization, Resources, Supervision, Funding acquisition, Validation, Investigation, Methodology, Project administration, Writing – review and editing

### Author ORCIDs

Sumedha Dahal (D) http://orcid.org/0000-0003-3682-5656
Sathees C Raghavan (D) http://orcid.org/0000-0003-3003-1417

### Ethics

This study was performed in strict accordance with the recommendations in the Guide for the Care and Use of Laboratory Animals of the Indian National Law on animal care and use. All of the animals were handled according to approved institutional animal care and use committee

protocols (CAF-SOP) of the Indian Institute of Science, Bangalore. The protocol was approved by the Committee on the Ethics of Animal Experiments of the Central Animal Facility (CAF/Ethics/526/2016).

### Decision letter and Author response
Decision letter https://doi.org/10.7554/eLife.69916.sa1
Author response https://doi.org/10.7554/eLife.69916.sa2

---

## Additional files

### Supplementary files
• Transparent reporting form

### Data availability
All data generated or analysed during this study are included in the manuscript and supporting files. Source data for each data is provided along with the figures.

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
