## [Editor Report]

This manuscript is of interest to researchers in the field of mitochondrial genome stability and mitochondrial genetic diseases and reports valuable findings that convincingly demonstrate that endonuclease G preferentially binds to mitochondrial genome regions which have a potential for forming G4 tetraplexes and induces DNA breaks that may lead to a common 9 bp deletion in the mitochondrial genome.

---

## [Decision Letter]

**Decision letter after peer review:**

Thank you for submitting your article "Unleashing a Novel Function of Endonuclease G in Mitochondrial Genome Instability" for consideration by *eLife*. Your article has been reviewed by 3 peer reviewers, including Wolf-Dietrich Heyer as the Reviewing Editor and Reviewer #1, and the evaluation has been overseen by Jessica Tyler as the Senior Editor. The following individual involved in review of your submission has agreed to reveal their identity: Sergei Mirkin (Reviewer #3).

Essential revisions:

The manuscript makes interesting observations that should advance the field of mitochondrial DNA metabolism and will be off interest for researchers in the field of mitochondrial genome stability and mitochondrial diseases.

The revision that is needed to make this manuscript, as submitted, publishable is too extensive. If the authors decide to address the comments by the additional experimentation needed, they are invited to resubmit a revised manuscript as a new submission that I would be happy to handle as BRE and seek to have the same reviewers evaluate the new submission.

Alternatively, the authors may want to consider tightening the manuscript and omit the reconstitution studies making sure that their interpretations are supported by the data and consider alternative mechanisms, when necessary. This more focused version would still contain sufficient new insights and allow the authors to further develop the reconstitution experiments with the required exhaustive separate incubation controls for a future publication.

1) Independent of specific criticism towards individual experiments or experimental tools, the manuscript is difficult to read, poorly assembled and would need a major overhaul in the text and figures. The text requires careful editing, there are many disagreements, and many cases where referrals such as 'its' or 'this' are unclear. The figures should be labeled that the experiments can be understood independent of the legend or method description. A few examples are given here.

• Figure 1A: The sequences of C1 and G1 are needed here. Giving more is confusing, as they do not relate to the figure. In Figure S1, all oligo sequences should be listed.

• Figure S1A: Define -/+ KCl under gel in legend.

• Figure S1B: Indicate inverted repeat sequences.

• Figure S3C: Where are the gel data? What is shown here? The description in the legend does not allow to understand the figure, what is the difference between the 3 lanes that are labeled as +.

• Figure S3: Unclear what the Westernblots show in D, F, and H.

• Figure S5F: Define the lanes. What is loaded in lane 4?

• Figure 4D: Label upper input and lower pull-down, also Figure 2G.

2) A key problem in the manuscript is that the authors mix their interpretation into the description of the results. A few examples are given here, but the authors should check the entire manuscript and correct this issue. The authors should carefully separate description of the results from their interpretation and refrain from overinterpreting their data (see also below). There is no need to overstate the result; exceptions or unexpected results must be discussed and acknowledged. For example:

• Figure 2F/G: What happened with CR7? Low threshold value but no PCR product? This is not described in the text.

• Figure 3D: The upper cleavage site stays the same, unlike what is described in the text.

• Figure S3A, B: It is unclear which band is EndoG. Is the Westernblot aligned with material loaded in slots 4-7 of the protein-stained gel? If so, why does the Westernblot intensity not reflect the staining intensity?

• Figure 3F: The Mt extract does only show the major EndoG sites but not the minor sites, which add up to a significant proportion of the products.

• Page 10, line 10: The mutant plasmid still has one cleavage site, cleavage is not abrogated as stated in the text.

• Page 10, line 22: Figure 3D is extract not purified EndoG as stated in line 22 of page 10.

• Figure 4E and F: What is the difference between both experiments? E: extract, F: cells? This could be clarified in the figure and text.

• Figure S5G: The description of the P1 results is not precise. There is clear protection of the C strand by EndoG in the upper third of the ladder.

• Figure 7A requires more description and quantitation. Figure 7 B/E: Why is the DAPI signal so variable? Figure 7G: the difference between lanes 1-3 and 4-6 should become apparent from the figure. For Figure 7G and H, only menadione data are shown, there are no IR data as stated in the text.

3) The BG4 antibody is problematic. Page 8, line 22: Why is there a BG4 signal in the cytoplasm in rho (0) cells? Is this also seen in rho (+) cells?

Also, the authors may want to discuss in more details image modification via Coste's mask. I can clearly see yellow dots in merged images for rho (0) cells, but somehow they are not converting to the white dotes when the Coste's mask is applied.

Figure 2 and 4: There are many BG4 signals? Do the signals all represent G4 tetraplex DNA? How can the authors ascertain the specificity and usefulness of this reagent?

The control in Figure 4A/B needs to be defined. What would be expectation from random overlap in this experiment given that both signals are very abundant?

4) The microscopy approach is less informative and overinterpreted. The level of resolution obtained with the confocal microscope approach used does not allow a conclusion on the localization beyond saying that the protein is present in the mitochondria and is not suitable for distinguishing the different mitochondrial compartments. As for the fractionation procedure used, I am not sure that it allows the separation of the matrix fraction from the intermembrane compartment. Furthermore, the description of the results displayed is not clear and there seem to be some mistakes in the labelling of the figures, making it harder to draw clear conclusions.

5) The mutant substrates are an excellent control for testing the G4 tetraplex hypothesis. What is the experimental evidence that these mutant oligonucleotides do not form tetraplexes?

6) The attempt to reconstruct the deletion formation in vitro is welcomed. However, these experiments lack a crucial control. In Figure 5B, there is product in the negative control? This raises the concern that bands labeled MMEJ products are PCR artifacts. The authors must conduct a separate incubation control by incubating both oligos separately with extracts, extract the DNA, mix, and then perform PCR detection to ascertain that bands labeled as products are formed during the extract incubation and not PCR artifacts. The recombination literature is replete with this artifact, and it must be eliminated here.

The separate incubation controls are also needed for the experiments with mt plasmid DNA and mitochondrial genomics DNA in Figure 5F and Figure S6.

In their exploration of the mechanisms leading to deletions, the data presented supports their model invoking MMEJ, at least for the in vitro system as shown by the sequencing of the deletions obtained either with the plasmid or with mitochondrial DNA. When mitochondrial DNA was used, however, it would seem that mutations in that region accumulate but they are not associated with deletions, which are proportionally less abundant than in the plasmid system. This finding merits more discussion.

The authors hypothesize that MMEJ repair of the DNA break caused by the endonuclease G produces the 9 bp-deletion. If this is the case, why just one out of four repeats is always deleted? Further, the authors may want to tone down their MMEJ model, since no direct evidence for this mechanism is provided. Other mechanisms, such as nick-repair can potentially lead to the same outcome.

7) Page 13, lines 20/21: Give the exact numbers for 'few' and 'many others'.

8) The EndoG nuclease experiments and conclusion that EndoG is responsible for the observed cleavages, would be strengthened by purifying a catalytic mutant of EndoG and showing dependance of cleavage on the EndoG active site. This is a standard control for the nuclease field.

9) The discussion is not well organized and should include an evaluation what the possible physiological function of the observation of EndoG release upon oxidative stress and deletion formation might be. Are these off-pathway accidents?

Is there a role of EndoG on G4 RNA tetraplexes? This refers also to the point about the BG4 antibody; does it recognize G4 RNA?

[Editors' note: further revisions were suggested prior to acceptance, as described below.]

Thank you for resubmitting your work entitled "Unleashing a Novel Function of Endonuclease G in Mitochondrial Genome Instability" for further consideration by *eLife*. Your revised article has been evaluated by Jessica Tyler (Senior Editor) and a Reviewing Editor.

The manuscript has been improved but there are some remaining issues that need to be addressed, as outlined below:

The manuscript makes interesting observations that should advance the field of mitochondrial DNA metabolism and will be off interest for researchers in the field of mitochondrial genome stability and mitochondrial diseases.

The revised manuscript is much improved, although there are number of points that should be addressed in a final revision.

Essential revisions:

1) The manuscript still requires some language editing.

2) Page 6, line 33: Overstatement, there are some pause site w/o KCl; change to significantly fewer sites and less pausing.

3) Figure 2 D: Primer DI12 is too inefficient to be meaningful. A new primer would be needed to make the point. The question is whether these data are needed in light of the mutant control?

4) Page 10, line 24-27 and Figure 6E: Make an additional mutant to test the prediction of the origin of the second remaining band.

5) Some points concerning the localisation of the protein in the mitochondria sub-fractions and its proposed recruitment to the matrix are still not convincing (please see specific comments below). There are enough novel data to justify its publication even without including the above-mentioned aspects, or if they are only presented as preliminary data to propose a hypothesis. The discussion should be amended to reflect that.

• Line 372: The fact that EndoG is present in mitochondria was already documented by several papers. It is not clear where the novelty is in this Results section. Furthermore, the title mentions TFAM as an inner membrane protein, which is not the case. As mentioned in the text, although TFAM is found associated with the membrane, it is a matrix protein. The low level of co-localisation and the limited resolution of the microscope used make do not allow to conclude much beyond the already known presence of Endo G in mitochondria. More detailed localisation studies would require electron microscopy or super resolution microscopy.

• Line 396: Based on the same arguments used above, the results described in the first paragraph of this section do not add much. The resolution of the images does not allow to conclude the presence of Endo G at G-quadruplexes through the co-localisation with the BG4 antibody. However, the PCR experiments provide evidence for that.

• The sub-mitochondrial fractionation described does not correspond to the references cited. In both papers cited there is a first step is a hypotonic shock that would yield a pellet of mitoplasts that can then be separated into inner membrane and matrix fractions. Here the fractionation is directly done in high salt and, to my understanding there is no an intermediate step allowing to get rid of the intermembrane fraction or the outer membrane.

There is all along an ambiguity regarding the localisation of EndoG. As mentioned by the authors the protein is normally found in the intermembrane space, therefore, the majority of it should be found in the soluble fraction with their fractionation protocol or lost if they use the hypotonic one. However, they find a substantial fraction in the membrane (inner or total?) fraction and this fraction does not decrease upon treatment while the soluble does. The question is then, where does the newly soluble protein come from?

[Editors' note: further revisions were suggested prior to acceptance, as described below.]

Thank you for resubmitting your work entitled "Unleashing a Novel Function of Endonuclease G in Mitochondrial Genome Instability" for further consideration by *eLife*. Your revised article has been evaluated by Jessica Tyler (Senior Editor) and a Reviewing Editor.

The manuscript has been improved but there are some remaining issues that need to be addressed, as outlined below:

1) There is still a contradiction in the section describing the localisation of EndoG within the mitochondria. The title of that section (page 11) is "Endonuclease G is expressed in mitochondria and colocalises with inner mitochondria membrane protein TFAM within the cells". However, based on the experiments described in the text that follows and shown in Figure 8, the authors conclude that it colocalises with TFAM which they define as "a transcription factor known to be present in the mitochondrial matrix".

2) As described the protocol for mitochondrial fractionation still does not provide a separation of the inner membrane from the outer one, nor does it separate the intermembrane space from the matrix (unless the description omitted a centrifugation step after the hypotonic shock). If this is the case the authors should be more careful in their statements and mention in the text on page 13, as they did in Figure 10, that what they are looking at is the partial release from the membrane to the soluble fractions.

---

## [Author Response]

Essential revisions:The manuscript makes interesting observations that should advance the field of mitochondrial DNA metabolism and will be off interest for researchers in the field of mitochondrial genome stability and mitochondrial diseases.The revision that is needed to make this manuscript, as submitted, publishable is too extensive. If the authors decide to address the comments by the additional experimentation needed, they are invited to resubmit a revised manuscript as a new submission that I would be happy to handle as BRE and seek to have the same reviewers evaluate the new submission.Alternatively, the authors may want to consider tightening the manuscript and omit the reconstitution studies making sure that their interpretations are supported by the data and consider alternative mechanisms, when necessary. This more focused version would still contain sufficient new insights and allow the authors to further develop the reconstitution experiments with the required exhaustive separate incubation controls for a future publication.

We thank the editors and reviewers for their positive comments and suggestions on our manuscript. We have omitted reconstitution experiments as suggested and tightened the manuscript by reorganizing the figures and text to address all the comments, as detailed below.

1) Independent of specific criticism towards individual experiments or experimental tools, the manuscript is difficult to read, poorly assembled and would need a major overhaul in the text and figures. The text requires careful editing, there are many disagreements, and many cases where referrals such as 'its' or 'this' are unclear. The figures should be labeled that the experiments can be understood independent of the legend or method description. A few examples are given here.• Figure 1A: The sequences of C1 and G1 are needed here. Giving more is confusing, as they do not relate to the figure. In Figure S1, all oligo sequences should be listed.

As suggested by the reviewer, we have removed the oligomer list of mutants from Figure 1 (now named as Figure 1B) and moved the mutants to Supplement Figure 1 (Figure 1—figure supplement 1B and Supplement Table 1).

• Figure S1A: Define -/+ KCl under gel in legend.

The suggested label is mentioned below the gel and added also to Figure legend (new Figure 1C and Figure 1—figure supplement 1C) and (page 33, lines 5-9 as well as page 40, lines 10-12).

• Figure S1B: Indicate inverted repeat sequences.

As suggested by the reviewer, we have now indicated the inverted repeat sequences (new Figure 1A and Figure 2—figure supplement 1).

• Figure S3C: Where are the gel data? What is shown here? The description in the legend does not allow to understand the figure, what is the difference between the 3 lanes that are labeled as +.

We are extremely sorry for the confusion. The gel data and the quantitation are now presented together in the same figure to avoid confusion (New Figure 6E, F). The 3 lanes labelled as + are the triplicate samples, which is now explained in the legend (page 36, lines 5-10).

• Figure S3: Unclear what the Westernblots show in D, F, and H.

The western blots in D, F, H confirm the identity of purified proteins in the fractions used (New Figure 7—figure supplement 1). Details are added to the legend (Page 41, lines 16-26).

• Figure S5F: Define the lanes. What is loaded in lane 4?

In old figure S5F (New Figure 9—figure supplement 2A), Lane 1 is marker, lanes 2 and 3 are pull down samples loaded after incubation of secondary antibody and anti-endonuclease G, respectively with mitochondrial extract. Lane 4 is mitochondrial extract.

• Figure 4D: Label upper input and lower pull-down, also Figure 2G.

As suggested, we have labelled the figures (new Figures 5C, 9D, Figure 9—figure supplement 1D).

2) A key problem in the manuscript is that the authors mix their interpretation into the description of the results. A few examples are given here, but the authors should check the entire manuscript and correct this issue. The authors should carefully separate description of the results from their interpretation and refrain from overinterpreting their data (see also below). There is no need to overstate the result; exceptions or unexpected results must be discussed and acknowledged. For example:• Figure 2F/G: What happened with CR7? Low threshold value but no PCR product? This is not described in the text.

When sequence of the region spanning CR7 was analysed for potential G4 motifs, we observed sequence that may support formation of 2 plate G4 structure. This could explain the observed binding of BG4 during semiquantitative PCR. However, the threshold value was based on real time PCR which is highly sensitive when compared to semiquantitative PCR. This could be the reason for absence of a band in the CR7. This is also described in the text now (Page 9, lines 11–18).

• Figure 3D: The upper cleavage site stays the same, unlike what is described in the text.

The mitochondrial G-quadruplex region I has the ability to form G-quadruplexes in multiple forms (Dahal et al., 2022). Mutating two of the sequences will still leave with 3-G stretches and 3 GNG sequences which makes the region, a ligand for Endonuclease G binding and cleavage (Figure 6—figure supplement 1). This is described in the main text (Page 11, lines 14-20).

• Figure S3A, B: It is unclear which band is EndoG. Is the Westernblot aligned with material loaded in slots 4-7 of the protein-stained gel? If so, why does the Westernblot intensity not reflect the staining intensity?

We have repeated the experiment and provided a new western blot for showing the presence of Endo G in the purified protein. We have marked the band representing Endo G in the figure. For both Endo G and its mutant, fractions 4 and 5, respectively were used for blotting (New Figure 7—figure supplement 1A-C).

• Figure 3F: The Mt extract does only show the major EndoG sites but not the minor sites, which add up to a significant proportion of the products.

Incubation with purified Endonuclease G resulted in cleavage at multiple positions across the G4 motif. It may be that multiple molecules of purified Endonuclease G may bind to different forms of G4 DNA resulting in cleavage at different positions. In the case of mitochondrial extract, there may be additional proteins, which can bind to G4 DNA, thus preventing cleavage at multiple sites. This has been now added in Page 11, lines 14-20.

• Page 10, line 10: The mutant plasmid still has one cleavage site, cleavage is not abrogated as stated in the text.

Although, 2 of the G-stretches are mutated, Region I still can fold into a G-quadruplex, thus one of the cleavage sites is not abrogated. We have corrected the statement in the mentioned paragraph (Page 10, line 22-26).

• Page 10, line 22: Figure 3D is extract not purified EndoG as stated in line 22 of page 10.

We apologize for the mistake. We have now corrected the statement (Page 10, lines 5-6, New figure 6).

• Figure 4E and F: What is the difference between both experiments? E: extract, F: cells? This could be clarified in the figure and text.

We apologise for the confusion. Figure 4E (New Figure 9E) corresponds to the use of mitochondrial extract and 4F (New Figure 9F) refers use of cells. The same is included in the text as well (page 13, lines 9-19).

• Figure S5G: The description of the P1 results is not precise. There is clear protection of the C strand by EndoG in the upper third of the ladder.

As suggested we have edited the text to match with the results in the Figure (New Figure 9—figure supplement 2B) (page 13, lines 20-25).

• Figure 7A requires more description and quantitation. Figure 7 B/E: Why is the DAPI signal so variable? Figure 7G: the difference between lanes 1-3 and 4-6 should become apparent from the figure. For Figure 7G and H, only menadione data are shown, there are no IR data as stated in the text.

We have now added description and quantitated figure 10A-D. We have also changed the images in figure 10B/E and the DAPI signal is now comparable between the panels. We apologise for the mistake. As mentioned, figure 10G and H is only for Menadione and not for IR.

3) The BG4 antibody is problematic. Page 8, line 22: Why is there a BG4 signal in the cytoplasm in rho (0) cells? Is this also seen in rho (+) cells?

BG4 binds to G-quadruplexes of nuclear DNA and RNA in cytoplasm (Biffi et al., 2014; Biffi et al., 2013). All the BG4 outside nucleus is not only because of G4 DNA in mitochondria, but also due to RNA G4 in the cytoplasm (Biffi et al., 2014). We have used mitotracker to indicate the mitochondria. Recently, we have also extensively investigated the specificity of BG4 towards G4 DNA (Javadekar et al., 2020). The BG4 signal in the cytoplasm is also seen in rho (+) cells.

Also, the authors may want to discuss in more details image modification via Coste's mask. I can clearly see yellow dots in merged images for rho (0) cells, but somehow they are not converting to the white dotes when the Coste's mask is applied.

We have now replaced the images to take care of the reviewer’s comments (Figure 4D).

Figure 2 and 4: There are many BG4 signals? Do the signals all represent G4 tetraplex DNA? How can the authors ascertain the specificity and usefulness of this reagent?The control in Figure 4A/B needs to be defined. What would be expectation from random overlap in this experiment given that both signals are very abundant?

Approximately 7,00,000 canonical G4 forming motifs of 25-35 nt are reported in the human nuclear genome (Chambers et al., 2015). Besides, recently 2 plate G-quadruplexes and involvement of GNG motifs were reported during G-quadruplex structure formation (Biffi et al., 2013; Das et al., 2016). If one considers all these into account, ~2% of the genome may support the formation of G-quadruplexes in human genome. Thus, the number of BG4 signals can be justified in the nucleus. Within mitochondria 5 canonical G4 motifs (Dahal et al., 2022) and >200 number of 2 plate G-quadruplexes (Brázda et al., 2019) have been reported. Considering that several mtDNA are present in one mitochondrion and several mitochondria are present in one cell, impact can be several fold. In addition to that RNA G4 structures have also been reported in cytoplasm (Biffi et al., 2014; Biffi et al., 2013).

The specificity and usefulness of the reagent has been extensively shown in the previous studies (Biffi et al., 2013; Das et al., 2016; Javadekar et al., 2020)

4) The microscopy approach is less informative and overinterpreted. The level of resolution obtained with the confocal microscope approach used does not allow a conclusion on the localization beyond saying that the protein is present in the mitochondria and is not suitable for distinguishing the different mitochondrial compartments.

We agree to the reviewer’s comments that the conclusion on localization based on microscopy approach is not preferred, and thus apart from microscopy we have used the fractionation of mitochondrial compartments to support the data (new Figure 10G, H).

As for the fractionation procedure used, I am not sure that it allows the separation of the matrix fraction from the intermembrane compartment. Furthermore, the description of the results displayed is not clear and there seem to be some mistakes in the labelling of the figures, making it harder to draw clear conclusions.

The fractionation procedure has been used as described in the previous literature (Sinha et al., 2010; Suzuki et al., 2002). We apologise for the mistakes in labelling and now it has been corrected and presented (new Figure 10).

5) The mutant substrates are an excellent control for testing the G4 tetraplex hypothesis. What is the experimental evidence that these mutant oligonucleotides do not form tetraplexes?

The mutant substrates have been used for the study to understand their mobility along with the wild type oligomers in the gel shift assays (Figure 1—figure supplement 1B, C). Apart from this, circular dichroism was performed and shown previously for this region and its mutants that they do not fold into G4 DNA (Dahal et al., 2022). Besides, we have also investigated the same region (with and without mutation) after cloning in a plasmid (new Figure 2).

6) The attempt to reconstruct the deletion formation in vitro is welcomed. However, these experiments lack a crucial control. In Figure 5B, there is product in the negative control? This raises the concern that bands labeled MMEJ products are PCR artifacts. The authors must conduct a separate incubation control by incubating both oligos separately with extracts, extract the DNA, mix, and then perform PCR detection to ascertain that bands labeled as products are formed during the extract incubation and not PCR artifacts. The recombination literature is replete with this artifact, and it must be eliminated here.The separate incubation controls are also needed for the experiments with mt plasmid DNA and mitochondrial genomics DNA in Figure 5F and Figure S6.In their exploration of the mechanisms leading to deletions, the data presented supports their model invoking MMEJ, at least for the in vitro system as shown by the sequencing of the deletions obtained either with the plasmid or with mitochondrial DNA. When mitochondrial DNA was used, however, it would seem that mutations in that region accumulate but they are not associated with deletions, which are proportionally less abundant than in the plasmid system. This finding merits more discussion.The authors hypothesize that MMEJ repair of the DNA break caused by the endonuclease G produces the 9 bp-deletion. If this is the case, why just one out of four repeats is always deleted? Further, the authors may want to tone down their MMEJ model, since no direct evidence for this mechanism is provided. Other mechanisms, such as nick-repair can potentially lead to the same outcome.

As discussed above, based on the suggestion by editors and reviewers we have now removed reconstitution studies from the manuscript. All suggested controls will be performed and a separate manuscript will be submitted at a later timepoint. We sincerely thank the editors for the constructive criticism and for the suggestion.

7) Page 13, lines 20/21: Give the exact numbers for 'few' and 'many others'.

We have now removed this figure from the manuscript completely and edited the text accordingly.

8) The EndoG nuclease experiments and conclusion that EndoG is responsible for the observed cleavages, would be strengthened by purifying a catalytic mutant of EndoG and showing dependance of cleavage on the EndoG active site. This is a standard control for the nuclease field.

We thank the editor and reviewer for the suggestion. We have now purified the catalytic mutant of Endo G and shown the cleavage pattern when the mutant protein is used (new Figure 7D, E).

9) The discussion is not well organized and should include an evaluation what the possible physiological function of the observation of EndoG release upon oxidative stress and deletion formation might be. Are these off-pathway accidents?

­­­­­As suggested, we have now reorganized the discussion. The observation that EndoG is released into matrix upon stress is new in the field. Although our data suggest that it may play a role in deletion formation, further studies are required to demonstrate this (Page 16, lines 13-17; page 17, lines 15-20).

Is there a role of EndoG on G4 RNA tetraplexes? This refers also to the point about the BG4 antibody; does it recognize G4 RNA?

Previously, it has been reported that BG4 binds to G4 RNA (Biffi et al., 2014; Biffi et al., 2013; Brázda et al., 2019). However, affinity of Endo G on G4 RNA tetraplexes is yet to be investigated. A detailed study is required before we can comment on this aspect.

[Editors' note: further revisions were suggested prior to acceptance, as described below.]

The manuscript has been improved but there are some remaining issues that need to be addressed, as outlined below:The manuscript makes interesting observations that should advance the field of mitochondrial DNA metabolism and will be off interest for researchers in the field of mitochondrial genome stability and mitochondrial diseases.The revised manuscript is much improved, although there are number of points that should be addressed in a final revision.

We thank the editors for their overall positive comments on our manuscript. Based on their suggestions following changes have been incorporated into the manuscript as detailed below.

Essential revisions:1) The manuscript still requires some language editing.

We have gone through the manuscript and made additional language corrections as suggested.

2) Page 6, line 33: Overstatement, there are some pause site w/o KCl; change to significantly fewer sites and less pausing.

As suggested, we have modified the text (Page 6, line 33-35).

3) Figure 2 D: Primer DI12 is too inefficient to be meaningful. A new primer would be needed to make the point. The question is whether these data are needed in light of the mutant control?

As rightly pointed out by the reviewer, the efficacy of primer (DI12) binding is inefficient, and this data does not add up significantly to the overall content of Figure 2. Thus, we have removed the panel from figure 2 and reorganised the rest of the figure (Revised Figure 2).

4) Page 10, line 24-27 and Figure 6E: Make an additional mutant to test the prediction of the origin of the second remaining band.

As suggested by the reviewer, we have now generated a new mutant to test the predicted basis of the second band and presented it as Figure 6-supplement figure 1. Results showed that upon generation of additional mutants, the second band, due to cleavage, disappeared. This further confirms the involvement of G4 DNA during the observed nicking (Page 10, lines 23-26).

5) Some points concerning the localisation of the protein in the mitochondria sub-fractions and its proposed recruitment to the matrix are still not convincing (please see specific comments below). There are enough novel data to justify its publication even without including the above-mentioned aspects, or if they are only presented as preliminary data to propose a hypothesis. The discussion should be amended to reflect that.

We agree with the reviewer, and as suggested, the immunolocalisation data showing the recruitment of Endonuclease G to the matrix is now removed from Figure 10. However, we have maintained sub-fractionation data as a preliminary observation (Figure 10).

• Line 372: The fact that EndoG is present in mitochondria was already documented by several papers. It is not clear where the novelty is in this Results section. Furthermore, the title mentions TFAM as an inner membrane protein, which is not the case. As mentioned in the text, although TFAM is found associated with the membrane, it is a matrix protein. The low level of co-localisation and the limited resolution of the microscope used make do not allow to conclude much beyond the already known presence of Endo G in mitochondria. More detailed localisation studies would require electron microscopy or super resolution microscopy.

We agree with the reviewer’s comments that the conclusion on sub-localisation based on the standard microscopy technique may not be sufficient to make the point. Thus, we have removed the immunofluorescence figures related to the localisation of EndoG to different mitochondrial compartments.

• Line 396: Based on the same arguments used above, the results described in the first paragraph of this section do not add much. The resolution of the images does not allow to conclude the presence of Endo G at G-quadruplexes through the co-localisation with the BG4 antibody. However, the PCR experiments provide evidence for that.

We thank the reviewer for their comment. We agree that the results described do not add much and the presence of BG4 cannot be concluded based on only colocalization with BG4. However, as stated, since we have other experiments (PCR-based ChIP) to evaluate the same question, we have maintained the data in the manuscript (Figures 4 and 9A).

• The sub-mitochondrial fractionation described does not correspond to the references cited. In both papers cited there is a first step is a hypotonic shock that would yield a pellet of mitoplasts that can then be separated into inner membrane and matrix fractions. Here the fractionation is directly done in high salt and, to my understanding there is no an intermediate step allowing to get rid of the intermembrane fraction or the outer membrane.

We apologise for not providing a detailed protocol. As mentioned in Sinha et al., we first performed the hypotonic shock to yield mitoplasts and separated them into inner membrane and matrix fractions (Sinha et al., 2010, Hum Mol Genet, vol 19, page 1248-1262; Suzuki et al., 2002, J Cell Sci, vol 115, pages 1895-1905). Accordingly, we have rewritten the section by providing sufficient detail to avoid ambiguity (Page 20, lines 24-32).

There is all along an ambiguity regarding the localisation of EndoG. As mentioned by the authors the protein is normally found in the intermembrane space, therefore, the majority of it should be found in the soluble fraction with their fractionation protocol or lost if they use the hypotonic one. However, they find a substantial fraction in the membrane (inner or total?) fraction and this fraction does not decrease upon treatment while the soluble does. The question is then, where does the newly soluble protein come from?

We agree with the reviewer that there has been ambiguity in the sublocalization of EndoG. While there are few papers suggesting the localization to inner membrane (David et al., 2006 Cell Death Differ;13(7):1147-55,Ishihara and Shimamoto (2006) JBC; 281, (10): 6726 –6733) others suggest its presence in the inner membrane space (Ohsato et al., 2002 Eur. J. Biochem.269, 5765–5770, Zhou et al., 2016 Science 353 (6297):394-9, Davies et al., 2003 Nucleic acids Res. 31(4): 1364–1373). With the protocol that we have used (the hypotonic method), if it was present only in the inner membrane space, Endo G would have been completely lost. However, further investigation is required to comment on this aspect.

We apologise for the ambiguity of quantification. We have now re-quantified the western blots and presented it as Figure 10B.

[Editors' note: further revisions were suggested prior to acceptance, as described below.]

The manuscript has been improved but there are some remaining issues that need to be addressed, as outlined below:1) There is still a contradiction in the section describing the localisation of EndoG within the mitochondria. The title of that section (page 11) is "Endonuclease G is expressed in mitochondria and colocalises with inner mitochondria membrane protein TFAM within the cells". However, based on the experiments described in the text that follows and shown in Figure 8, the authors conclude that it colocalises with TFAM which they define as "a transcription factor known to be present in the mitochondrial matrix".

We thank the reviewer for pointing out this and apologize for the error. We have now corrected the title section on page 12, lines 1, 2.

2) As described the protocol for mitochondrial fractionation still does not provide a separation of the inner membrane from the outer one, nor does it separate the intermembrane space from the matrix (unless the description omitted a centrifugation step after the hypotonic shock). If this is the case the authors should be more careful in their statements and mention in the text on page 13, as they did in Figure 10, that what they are looking at is the partial release from the membrane to the soluble fractions.

Thank you for the suggestion. We have now edited the text to avoid the discrepancy (page 13, line 31).